# Private Rate-Constrained Optimization with Applications to Fair Learning

**Mohammad Yaghini**[*]
University of Toronto & Vector Institute
mohammad.yaghini@mail.utoronto.ca

**Tudor Cebere**[*]
PreMeDICaL team, Inria, Idesp, Inserm
Université de Montpellier
tudor.cebere@inria.fr

**Michael Menart**
University of Toronto & Vector Institute
menart.2@osu.edu

**Aurélien Bellet**
PreMeDICaL team, Inria, Idesp, Inserm
Université de Montpellier
aurelien.bellet@inria.fr

**Nicolas Papernot**
University of Toronto & Vector Institute & Inria
nicolas.papernot@utoronto.ca

## Abstract

Many problems in trustworthy ML can be expressed as constraints on prediction rates across subpopulations, including group fairness constraints (demographic parity, equalized odds, etc.). In this work, we study such constrained minimization problems under differential privacy (DP). Standard DP optimization techniques like DP-SGD rely on objectives that decompose over individual examples, enabling per-example gradient clipping and noise addition. Rate constraints, however, depend on aggregate statistics across groups, creating inter-sample dependencies that violate this decomposability. To address this, we develop RaCO-DP, a DP variant of Stochastic Gradient Descent-Ascent (SGDA) that solves the Lagrangian formulation of rate constraint problems. Through careful design, the extra privacy cost incurred by incorporating these constraints in our approach is limited to that of privately estimating a histogram over each mini-batch at every step. We prove the convergence of our algorithm through a novel analysis of SGDA that leverages the linear structure of the dual parameter. Empirical results[1] show that our method Pareto-dominates existing private learning approaches under group fairness constraints and also achieves strong privacy–utility–fairness performance on neural networks.

## 1 Introduction

From fair learning (Zafar et al., 2017; Goel et al., 2018; Agarwal et al., 2018; Cotter et al., 2019a) to robust optimization (Chen et al., 2017; Cotter et al., 2019b) and cost-sensitive learning (Mienye & Sun, 2021; Cotter et al., 2019b), many Machine Learning (ML) tasks can be formulated as constrained optimization problems. In such problems, the goal is to minimize the model's overall error subject to *rate constraints*, which enforce conditions on prediction rates across subsets of the training data. For instance, ensuring fairness in a resume screening system might require similar rates of positive outcomes (e.g., resumes selected for human review) across gender groups, a criterion known as equalized odds (Hardt et al., 2016). Similarly, in a medical setting, a decision-support system may need to strictly limit its false negative rate, e.g., to reduce the risk of misdiagnosing cancerous tumours as benign.

These learning tasks often require training models on sensitive data, such as employee or patient records in our examples. Publicly releasing these models can expose data owners to privacy attacks (Shokri et al., 2017) and disclose personal data without consent (Sweeney, 2002). Without

---

[*]Equal contribution
[1]Code is available at: https://github.com/cleverhans-lab/dp-raco

proper safeguards, these risks can harm individuals, undermine trust in AI systems, and discourage data sharing in critical applications like medical research. Despite recent advances, differentially private (DP) constrained optimization has focused almost exclusively on fairness constraints (Jagielski et al., 2019; Mozannar et al., 2020; Tran et al., 2023; Berrada et al., 2023; Lowy et al., 2023). Methods tailored to fairness do not extend to the broader family of rate constraints that arise in practice. We bridge this gap with the first general DP framework for arbitrary rate-constrained problems. Our approach expands DP's reach to previously incompatible applications and pushes the Pareto frontier of utility, privacy, and constraint satisfaction, including fairness, beyond the state of the art.

Differential Privacy (DP) is the standard framework for private data analysis that has been successfully applied to training and releasing *unconstrained* models with formal privacy guarantees. A key approach is the widely used DP-SGD algorithm (Song et al., 2013b; Bassily et al., 2014; Abadi et al., 2016), which ensures DP by clipping per-sample gradients and adding calibrated noise to the averaged gradient. This process bounds each data sample's influence. Incorporating rate constraints presents a challenge, however, because unlike typical training losses, these constraint functions (or their regularizer counterparts) *do not readily decompose into per-sample terms*. This fundamental incompatibility with per-sample processing makes it difficult to integrate them with standard DP-SGD and its variants. Our key contribution is the introduction of a DP optimization algorithm that overcomes this limitation, enabling private optimization subject to rate constraints.

To address the challenge of privately enforcing rate constraints, we propose *generalized rate constraints*. Generalized rate constraints allow us to express all rate constraints in a common form based on statistics (i.e., histograms) on *disjoint* subgroups within the dataset. This common structure is the key to our privacy solution: it allows us to efficiently gather all the necessary information to evaluate these constraints under DP. Beyond its advantages for privacy, our method offers greater flexibility compared to prior work (Cotter et al., 2019b) by extending to scenarios with multiple output classes.

With generalized rate constraints at hand, we introduce RaCO-DP, a framework for optimizing machine learning models under rate constraints with differential privacy (DP). RaCO-DP is a differentially private variant of the Stochastic Gradient Descent Ascent (SGDA) algorithm which leverages a Langrangian formulation and generalized rate constraints to overcome the decomposability obstacle. Our core insight exploits the structure provided by these constraints: we can efficiently compute differentially private statistics (e.g., histograms) for these subgroups. By privatizing these statistics at each step, we enable private evaluation of the constraint function, and its per-sample constraint gradients. We provide a formal convergence analysis of RaCO-DP, proving that even for non-convex optimization problems, our method converges to an approximate stationarity point. To achieve this, we introduce a novel approach to analyzing SGDA that accounts for bias in gradient estimates and exploits the linear structure of the dual update to enhance convergence speed.

As a concrete application, we study private learning under various constraints, including demographic parity (Dwork et al., 2012), false negative rate, and equalized odds. Our method achieves new state-of-the-art (SOTA) results on four standard benchmarks and Pareto-dominates the prior best approach (Lowy et al., 2023) in accuracy–fairness–privacy trade-offs. We further demonstrate that RaCO-DP scales beyond convex models, achieving strong performance for deep neural networks (ResNet16 He et al. (2016) on `CelebA` Liu et al. (2015)). We also validate its effectiveness on tasks involving multiple sensitive groups (demographic parity on the `ACSEmployment` dataset), scaling to 18 subgroup constraints. Additionally, our method offers two advantages over existing approaches explicitly designed for fairness constraints. First, it provides stronger privacy guarantees than prior approaches that only consider the privacy of the sensitive label (Jagielski et al., 2019; Tran et al., 2023), akin to label privacy (Ghazi et al., 2021). Second, it allows practitioners to *directly* specify the maximum disparity, unlike previous penalization-based methods, such as (Lowy et al., 2023), which offer only indirect control over the desired fairness level via hyperparameter tuning.

## 2 BACKGROUND

### 2.1 DIFFERENTIAL PRIVACY

Differential Privacy (DP) (Dwork et al., 2006) has become the de facto standard in privacy-preserving ML thanks to the robustness of its guarantees, its desirable behaviour under post-processing and composition, and its extensive algorithmic framework. We recall the definition below and refer to

Dwork & Roth (2014) for more details. Let $\mathcal{X}, \mathcal{Y}$ be the input and output spaces, respectively. We fix some $m \in \mathbb{N}$ and denote by $\mathcal{D}$ the space $(\mathcal{X} \times \mathcal{Y})^m$ of datasets of size $m$.

**Definition 2.1** (Differential Privacy). *A mechanism $\mathcal{M} : \mathcal{D} \to \mathcal{O}$ is $(\varepsilon, \delta)$-DP if for all datasets $D, D' \in \mathcal{D}$ differing in one datapoint and for all events $\mathcal{O}$: $P[\mathcal{M}(D) \in \mathcal{O}] \leq e^\varepsilon P[\mathcal{M}(D') \in \mathcal{O}] + \delta$.*

In the above definition, $\delta \in (0, 1)$ can be thought of as the failure probability, and $\varepsilon > 0$ the privacy loss; smaller $\epsilon$ and $\delta$ correspond to stronger privacy guarantees.

Differentially Private Stochastic Gradient Descent (DP-SGD) (Song et al., 2013b; Bassily et al., 2014; Abadi et al., 2016) serves as the foundational algorithm in private ML. Given a training dataset $D \in \mathcal{D}$ and model parameters $\theta \in \mathbb{R}^d$, DP-SGD aims to privately solve (in the sense of definition 2.1) the empirical risk minimization problem $\min_{\theta \in \mathbb{R}^d} \ell(\theta)$, where $\ell(\theta) = \frac{1}{|D|} \sum_{x \in D} \ell(\theta; x)$ and $\ell(\theta, \cdot)$ is a loss function differentiable in $\theta$. DP-SGD follows the standard SGD update, but guarantees differential privacy by (i) capping each data point's influence on the gradient through gradient clipping, and (ii) injecting Gaussian noise into the clipped gradients.

Each iteration $t \in [T]$ of DP-SGD incurs a privacy loss $(\varepsilon_t, \delta_t)$. Privacy composition (Dwork et al., 2006; Kairouz et al., 2015) and privacy accounting (Abadi et al., 2016; Gopi et al., 2021; Doroshenko et al., 2022) are techniques that aggregate these per-step privacy losses into a total privacy guarantee $(\varepsilon, \delta)$ that holds for the entire optimization process.

## 2.2 Constrained Optimization via Lagrangian

In this work, we aim to solve *constrained* empirical risk minimization problems of the form:

$$\min_{\theta \in \mathbb{R}^d} \left\{ \ell(\theta) := \frac{1}{|D|} \sum_{x \in D} \ell(\theta; x) \right\} \quad \text{s.t.} \quad \forall j \in [J],\ \Gamma_j(\theta) \leq \gamma_j. \tag{1}$$

where $\Gamma_j : \mathbb{R}^d \mapsto \mathbb{R}^+$ are the constraint functions and $\gamma \in (\mathbb{R}^+)^J$ are slack parameters. We focus on inequality constraints, as equality constraints are generally infeasible under differential privacy (Cummings et al., 2019).

Due to the difficulty of solving (1) directly, we instead solve an equivalent min-max optimization problem with respect to the Lagrangian:

$$\min_{\theta \in \mathbb{R}^d} \max_{\lambda \in \Lambda} \left\{ \mathcal{L}(\theta, \lambda) := \ell(\theta) + R(\theta, \lambda) \right\}, \quad \text{where: } R(\theta, \lambda) = \sum_{j=1}^J \lambda_j (\Gamma_j(\theta) - \gamma_j). \tag{2}$$

Here, $\lambda \in \Lambda \subseteq (\mathbb{R}^+)^J$ are the Lagrange multipliers, often referred to as the dual parameter, while $\theta$ is the primal parameter. One of the simplest algorithms to solve (2) is the generalization of (S)GD known as (Stochastic) Gradient Descent-Ascent, (S)GDA (Nemirovski et al., 2009).

**Definition 2.2** (GDA). *At each iteration $t$:*

$$\theta^{(t+1)} \leftarrow \theta^{(t)} - \eta_\theta \nabla_\theta \mathcal{L}(\theta^{(t)}, \lambda^{(t)}), \quad \text{and} \quad \lambda^{(t+1)} \leftarrow \Pi_\Lambda \left( \lambda^{(t)} + \eta_\lambda \nabla_\lambda \mathcal{L}(\theta^{(t)}, \lambda^{(t)}) \right). \tag{3}$$

*where $\eta_\theta$ and $\eta_\lambda$ are step sizes and $\Pi_\Lambda$ performs orthogonal projection onto $\Lambda$. The stochastic version (SGDA) replaces exact gradients with stochastic estimates.*

## 2.3 Rate Constraints

In this work, we focus on constraints that relate to prediction behavior over subsets of the dataset. Consider a model $h : \mathcal{X} \times \mathbb{R}^d \mapsto \mathbb{R}^K$ that maps inputs from feature space $\mathcal{X}$ to real-valued prediction scores over the label set $\mathcal{Y} = \{1, \ldots, K\}$ using parameters $\theta \in \mathbb{R}^d$. Formally, (hard) prediction rates count the number of points in a dataset $D$ for which the model predicts a certain label $k \in \mathcal{Y}$: $P_k^{\text{hard}}(D; \theta) = \frac{1}{|D|} \sum_{x \in D} \mathbb{1}_{[\arg\max_{k' \in [K]} \{h(\theta; x)_{k'}\} = k]}$.

As the indicator function is non-differentiable and thus challenging to optimize, we will use differentiable versions of these constraints. We rely on the tempered softmax function $\sigma_\tau(z)_k = \frac{\exp(-\tau z_k)}{\sum_{l=1}^K \exp(-\tau z_l)}$, where the temperature parameter $\tau \in \mathbb{R}^+$ controls the sharpness of the probability distribution. This allows us to define soft prediction rates:

$$P_k(D; \theta, \tau) = \frac{1}{|D|} \sum_{x \in D} \sigma_\tau(h(\theta; x))_k, \quad \text{for } k \in \mathcal{Y}. \tag{4}$$

Observe that $\lim_{\tau \to \infty} P_k(D; \theta, \tau) = P_k^{\text{hard}}(D; \theta)$.

*Rate constraints*, as defined by Goh et al. (2016) and Cotter et al. (2019b) for binary classification ($\mathcal{Y} = \{0, 1\}$), are linear combinations of a classifier's prediction rates across different data partitions:

$$\sum_{q \in [Q]} \alpha_q P_1(D_q, \theta) + \beta_q P_0(D_q, \theta) \leq \gamma, \tag{5}$$

$Q > 0$, $\alpha_q, \beta_q \in \mathbb{R}$ are mixing coefficients, $D_1, ..., D_Q \subseteq D$ and $\gamma$ is the slack parameter. Examples of rate constraints in this form include ensuring that the fraction of positive predictions across different demographic groups stays within a specified threshold, or requiring the model to achieve a minimum level of precision or recall for both classes (Cotter et al., 2019a).

Such constraints are limited to binary classification, and a multiclass generalization is not immediate. More critically, it is unclear how to evaluate rate constraints in (5) while efficiently preserving privacy. These are issues we address in the next section.

## 3 PRIVATE LEARNING WITH GENERALIZED RATE CONSTRAINTS

**Generalized rate constraints.** We propose a generalized form of rate constraints that (i) applies to the multi-class setting and (ii) exploits shared structure to enable accurate private estimation.

This shared structure is a partition $\{D_1, ..., D_Q\}$ of the dataset $D$ for some $Q > 0$, which we refer to as the "global" partition. We then allow each rate constraint to incorporate prediction rates over any recombination of sub-datasets in the global partition. This structure is flexible, as it allows each rate constraint to have its own "local" datasets, provided that each of these datasets can be formed as a union of sets from the global partition. For example, in the context of fairness constraints, the global partition corresponds to the the sensitive groups (e.g., Asian, Black, Caucasian), and local datasets for one of the constraint is {Asian, Non-Asian} where Non-Asian = {Black, Caucasian}. See Figure 1.

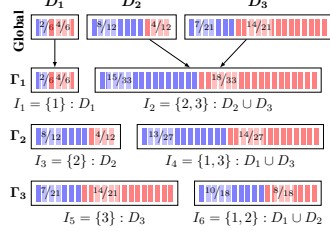

Figure 1: **Each rate constraint of the form** (6) **builds local datasets based on the global partition.** A class-1 (class-0) prediction is shown with a blue (red) square. Prediction rates $P_0, P_1$ are shown as fractions. As an example, let $D_1$, $D_2$, and $D_3$ be the set of Asian, Black, and Caucasian individuals in the dataset, respectively. Constraint $\Gamma_1$ builds its local datasets as $\{\{D_1\}, \{D_2 \cup D_3\}\}$, i.e. {Asian, Non-Asian}, from the global partition $\{D_1, D_2, D_3\}$ using the set of index subsets $\mathcal{I}_1 = \{I_1, I_2\}$.

Formally, given this global partition, we assume for each $j \in [J]$ there exists a family of subsets of $[Q]$, denoted $\mathcal{I}_j \subseteq 2^{[Q]}$, and a weight vector $\alpha_j \in \mathbb{R}^{|\mathcal{I}_j| \cdot K}$, such that the constraint $\Gamma_j$ can be written in the following form:[2]

$$\Gamma_j(\theta) = \sum_{I \in \mathcal{I}_j} \sum_{k=1}^{K} \alpha_{j,I,k} P_k(\cup_{i \in I} D_i; \theta). \tag{6}$$

Assuming such a global partition is not restrictive, as the trivial partition $D_q = x_q$ (with $Q = |D|$) can always be used. However, as Section 4 will show, a smaller partition size $Q$ enables a better privacy-utility trade-off through more effective noise use.

**Remark 1.** *For common rate constraints, the best global partition will be readily apparent (see the application to fairness below). The smallest global partition can however be defined explicitly. Let $\bar{D}_1, ..., \bar{D}_{\bar{Q}}$ be the (possibly non-disjoint) subsets of $D$ over which the functions $\Gamma_1, ..., \Gamma_J$ compute a prediction rate as per Eq. (6). Then the smallest global partition assigns any two data points $x$ and $x'$ in the same $D_q$ if and only if $\{\forall \bar{q} \in [\bar{Q}], \{x, x'\} \cap \bar{D}_{\bar{q}} \in \{\emptyset, \{x, x'\}\}\}$.*

Note that each rate constraint $\Gamma_j$ is uniquely defined by the subset family $\mathcal{I}_j$ and vector $\alpha_j$. Both parameters are public, specifying only the constraint's structure and containing no sensitive data.

**Application to fair learning.** Group fairness in machine learning aims to prevent models from making biased decisions across different sensitive groups. We show below our general form of rate constraints (4) allows to capture the popular group fairness notion of demographic parity (Barocas et al., 2023).

---

[2]With some abuse of notation, we denote by $\alpha_{j,I,k}$ the entry of $\alpha_j$ corresponding to subset $I$ and label $k$.

More generally, all common group fairness measures can be formulated as rate constraints (Cotter et al., 2019b). We provide details on formulating other fairness notions in Appendix A.

**Definition 3.1** (Demographic Parity). *Assume each feature vector $x \in \mathcal{X}$ contains a sensitive attribute, denoted as $Z$, taking on values in $\mathcal{Z} \subset \mathbb{Z}$. A classifier $h(\theta; \cdot)$ satisfies demographic parity w.r.t. $Z$ if $\Pr[\hat{Y} = k \mid Z = z] = \Pr[\hat{Y} = k]$ for all $z \in \mathcal{Z}, k \in \mathcal{Y}$, where $\hat{Y} = h(\theta; X)$.*

In practice, probabilities are replaced by empirical rates $P_k$. With slack $\gamma$, this gives

$$P_k(D[Z = z]; \theta) - P_k(D[Z \neq z]; \theta) \leq \gamma, \quad \forall z \in \mathcal{Z}, k \in \mathcal{Y}, \tag{7}$$

leading to $J = |\mathcal{Z}| \cdot |\mathcal{Y}|$ constraints of the form (6). The global partition has size $Q = |\mathcal{Z}|$ with elements $D_z = D[Z = z]$. For the constraint indexed by $(z, y) \in \mathcal{Z} \times \mathcal{Y}$, we take $\mathcal{I} = (\{z\}, [|\mathcal{Z}|] \setminus z)$ and define $\alpha_{\{z\},y} = 1$, $\alpha_{\{|\mathcal{Z}|\} \setminus z, y} = -1$, with all other components zero.

**Objective.** With these definitions in place, we can now state our objective: solve problem (2) with generalized rate constraints of the form (6) under DP.

## 4 RaCO-DP: Private Rate-Constrained Optimization

We introduce RaCO-DP (Algorithm 1), an algorithm for rate-constrained optimization that extends SGDA (Definition 2.2) to satisfy DP. Each iteration $t$ of RaCO-DP operates on a mini-batch $B^{(t)}$ and consists of three key components that we will describe in detail in this section:

1. **Private histogram computation:** For each class $k \in [K]$ and part $q \in [Q]$ of the partition, we privately estimate the sum of model predictions $\sigma(h(\theta_t; x))_k$ over the points $x$ in the mini-batch $B^{(t)}$ that belong to $D_q$, storing these counts in a histogram $H^{(t)}$ (Section 4.1).

2. **Private primal updates:** We derive per-sample gradients for the primal update of SGDA based on post-processing the histogram $H^{(t)}$. We then clip, average and privatize these gradients using the Gaussian mechanism (Section 4.2).

3. **Private dual updates:** We compute all constraint values by again post-processing the histogram $H^{(t)}$, allowing us to perform the dual update at no additional privacy cost (Section 4.3).

---

**Algorithm 1** RaCO-DP

**Require:** Dataset $D$, Parameter $\theta_0$, learning rates $\eta_\theta$, $\eta_\lambda$, Gaussian noise variance $\sigma$, Laplace parameter $b$, clipping norm $C$, sampling rate $r$, slack parameter vector $\gamma$, loss function $\ell(\theta; \cdot)$

1: Initialize $\lambda^{(0)} \leftarrow [0]$
2: **for** each $t \in \{0, \ldots, T - 1\}$ **do**
3:     $B^{(t)} \leftarrow \text{PoissonSample}(D, r)$
    *Private Histogram:*
4: 
5:     $\hat{H}_{q,k}^{(t)} \leftarrow \sum\limits_{x \in D_q \cap B^{(t)}} \sigma(h(\theta^{(t)}; x))_k + \text{Lap}(\frac{1}{b})$
    *Primal Update:*
6:     $Z^{(t)} \sim \text{Gaussian}(0, \mathbb{I}_d \sigma^2)$
7:     $\forall x \in B^{(t)}$ : set $g_{x,\theta}^{(t)}$ as (see also Eq.(10))
    $\frac{1}{r|D|} \nabla_\theta \ell(\theta^{(t)}; x) + \nabla_\theta \hat{R}(\theta^{(t)}, \lambda^{(t)}; \hat{H}^{(t)}, x)$
8:     $g_\theta^{(t)} \leftarrow \left( \sum_{x \in B^{(t)}} \text{clip}(g_{x,\theta}^{(t)}, \frac{C}{r|D|}) \right) + Z^{(t)}$
9:     $\theta^{(t+1)} \leftarrow \theta^{(t)} - \eta_\theta g_\theta^{(t)}$
    *Dual Update:*
10:     $[g_\lambda^{(t)}]_j \leftarrow \Gamma_j^{\text{post}}(\hat{H}^{(t)}) - \gamma_j, \quad \forall j \in [J]$
11:     $\lambda^{(t+1)} \leftarrow \Pi_\Lambda(\lambda^{(t)} + \eta_\lambda g_\lambda^{(t)})$ (see Eqn. (12)
12: **end for**
13: **return** $\theta^{(T)}$

---

### 4.1 Private Histogram Computation

Our algorithm's primal and dual updates compute prediction rates across dataset partition parts $D_1, \ldots, D_Q$ for each class. To track these prediction rates privately, we construct a histogram $H^{(t)} \in \mathbb{R}^{Q \times K}$ that counts (soft) model predictions for each combination of part $q$ and class $k$. For a given model parameter $\theta$, each sample $x_i$ in the mini-batch $B^{(t)}$ belongs to exactly one part $D_q$ of the partition but can influence the counts of all $K$ classes through the softmax probabilities $\sigma(h(\theta; x))_k$. This non-private histogram is constructed by accumulating these softmax vectors:

$$H_{q,k}^{(t)} = \sum\nolimits_{D_q \cap B^{(t)}} \sigma(h(\theta; x_i))_k. \tag{8}$$

To make this histogram differentially private, we use the Laplace mechanism (Dwork et al., 2006). The $\ell_1$ sensitivity of $H^{(t)}$ is 1, as each sample belongs to one element of the global partition and its softmax predictions sum to 1 across classes. Thus, we achieve $\varepsilon$-DP by adding Laplace noise to each element:

$$\hat{H}_{q,k}^{(t)} = H_{q,k}^{(t)} + \text{Lap}(1/\varepsilon), \quad \forall q \in [Q], k \in \mathcal{Y}. \tag{9}$$

**Remark 2.** *We focus on the Laplace mechanism for simplicity, but we note that our framework can readily accommodate other differentially private histogram mechanisms that may provide better utility in some regimes, e.g., when the histogram is high-dimensional and sparse (Wilkins et al., 2024).*

In the following sections, we will see that $\hat{H}^{(t)}$ contains all the necessary information to compute the quantities related to the rate constraints required for both the primal and dual updates, thereby avoiding additional privacy costs that would arise from composing multiple queries.

## 4.2 PRIVATELY COMPUTING THE PRIMAL GRADIENT

A key requirement in DP-SGDA is that each sample has a bounded contribution to gradient updates. To satisfy this in RaCO-DP, we decompose the Lagrangian into per-sample terms. While the loss $\ell(\theta)$ naturally decomposes as in standard DP-SGD, the regularizer $R$ is more challenging.

Given a mini-batch $B$, define $B_{\cap I} = B \cap (\underset{i \in I}{\cup} D_i)$ and recall that $\mathcal{I}_j$ and $\alpha_j \in \mathbb{R}^{|\mathcal{I}_j| \times K}$ denote the family of subsets and weight vector associated with constraint $\Gamma_j$. The minibatch-level regularizer is,

$$R(\theta, \lambda; B) = \sum_{j=1}^{J} \lambda_j \Big( \big( \sum_{I \in \mathcal{I}_j} \sum_{k}^{K} \alpha_{j,I,k} P_k(B_{\cap I}; \theta) \big) - \gamma_j \Big) = \sum_{j=1}^{J} \lambda_j \Big( \big( \sum_{I \in \mathcal{I}_j} \sum_{k}^{K} \sum_{x \in B_{\cap I}} \frac{\alpha_{j,I,k}}{|B_{\cap I}|} \sigma(h(\theta; x)_k) \big) - \gamma_j \Big).$$

Note that this may be a *biased* estimate of $R(\theta, \lambda)$ due to the normalization term $|B_{\cap I}|$. We account for this bias in our convergence analysis (Section 5).

We would like a per-sample decomposition of $R(\theta, \lambda; B)$. The main obstacle in such a decomposition are the quantities $|B_{\cap I}|$, which depend on the entire mini-batch. We overcome this by first noting that $|B_{\cap I}| = \sum_{i \in I} \sum_{k} H_{i,k}^{(t)}$, leading to the following per-sample regularizer estimator at point $x \in D$:

$$\hat{R}(\theta, \lambda; H, x) = \sum_{j=1}^{J} \lambda_j \Big( \big( \sum_{I \in \mathcal{I}_j} \sum_{k_1 \in K} \frac{\alpha_{j,I,k_1} \mathbb{1}_{[x \in B_{\cap I}]}}{\sum_{i \in I} \sum_{k_2 \in K} H_{i,k_2}^{(t)}} \sigma(h(\theta; x))_{k_1} \big) - \gamma_j \Big),$$

and thus the overall per-sample gradient is given by,

$$\frac{\nabla_\theta \ell(\theta; x)}{r|D|} + \sum_{j=1}^{J} \sum_{I \in \mathcal{I}_j} \sum_{k_1 \in [K]} \frac{\lambda_j \, \alpha_{j,I,k_1} \, \mathbb{1}_{[x \in B_{\cap I}]}}{\sum_{i \in I} \sum_{k_2 \in [K]} H_{i,k_2}^{(t)}} \nabla_\theta [\sigma(h(\theta; x))_{k_1}]. \tag{10}$$

Then, since $H$ depends on the mini-batch $B$, we use instead its differentially private version $\hat{H}$. Note that the normalizing term $\frac{1}{r|D|}$ is necessary to correctly implement clipping in Line 8 of Algorithm 1.

**Remark 3.** *For specific constraints, the estimation of $|B_{\cap I}|$ can be further refined. For example, when each $\mathcal{I}_j$ is itself a partition of $[Q]$, the sensitivity of $\sum_i \sum_{k_2} H_{i,k_2}^{(t)}$ is at most 1, and adding Laplace noise directly results in a tighter estimate of $|B_{\cap I}|$.*

With this per-sample decomposition, we can apply standard DP-SGD techniques: clipping per-sample gradients, averaging them over the mini-batch, and adding Gaussian noise to preserve privacy.

## 4.3 PRIVATELY COMPUTING THE CONSTRAINT DUAL GRADIENT

For each constraint $\Gamma_j$ and corresponding slack parameter $\gamma_j$, the gradient of the Lagrangian w.r.t $\lambda$ is:

$$\nabla_\lambda \mathcal{L}(\theta^{(t)}, \lambda^{(t)})_j = \Gamma_j(\theta^{(t)}; B^{(t)}) - \gamma_j, \tag{11}$$

The dual update in RaCO-DP thus requires evaluating rate constraints on the current mini-batch $B^{(t)}$, incurring a privacy cost. To avoid this additional cost, we introduce a post-processing function $\Gamma_j^{\text{post}} : \mathbb{R}^{Q \times K} \mapsto \mathbb{R}$ that operates directly on the private histogram $\hat{H}^{(t)}$. This function replaces each sum of model predictions $\sum_{i \in D_q \cap B^{(t)}} \sigma(h(\theta; x_i))_k$ with the corresponding histogram count $\hat{H}_{q,k}^{(t)}$:

$$\Gamma_j^{\text{post}}(\hat{H}^{(t)}) = \sum_{I \in \mathcal{I}_j} \sum_{k_1 \in [K]} \frac{\alpha_{j,I,k_1} \sum_{i \in I} \hat{H}_{i,k_1}^{(t)}}{\sum_{i \in I} \sum_{k_2 \in [K]} \hat{H}_{i,k_2}^{(t)}}. \tag{12}$$

As $\hat{H}^{(t)}$ is already DP, the post-processing property ensures this step incurs no additional privacy cost.

**Remark 4.** *As standard in private optimization, mini-batches $B^{(t)}$ are constructed via Poisson sampling, with each datapoint included with probability $r$. This allows us to leverage privacy amplification by subsampling (Kasiviswanathan et al., 2011; Balle et al., 2018) for the Laplace mechanism in histogram computation (Section 4.1) and the Gaussian mechanism for per-sample gradients (Section 4.2).*

RaCO-DP's efficiency relies on a private histogram $H^{(t)}$ enabling per-sample gradient computation and private constraint evaluation, key to handling rate constraints under DP. The privacy guarantees of Algorithm 1 follow from composing the subsampled Laplace and Gaussian mechanisms over $T$ steps.

**Theorem 4.1.** *Let $b \geq 2 \max \left\{ \frac{1}{\epsilon}, \frac{r\sqrt{T\log(T/\delta)}}{\epsilon} \right\}$ and $\sigma \geq 10 \max \left\{ \frac{C\log(T/\delta)}{r|D|\epsilon}, \frac{C\sqrt{T}\log(T/\delta)}{|D|\epsilon} \right\}$, then Algorithm 1 is $(\epsilon, \delta)$-DP.*

The proof is in Appendix B. We present this result primarily for to provide intuition about parameter scaling. In our experiments, we use a tighter privacy accountant that improves both constants and logarithmic factors. Additionally, in our convergence analysis, we offer a more detailed examination when the Lagrangian is Lipschitz and the algorithm is run without clipping.

## 5 CONVERGENCE AND UTILITY ANALYSIS

Consider the function defined as $\Phi(\theta) = \max_{\lambda \in \Lambda} \{ \mathcal{L}(\theta, \lambda) \}$. Ideally, one would want to show that Algorithm 1 approximately minimizes $\Phi$. However, due to the fact that $\Phi$ may be non-convex, finding an approximate minimizer is intractable in general. In fact, because the constraint functions, $\{\Gamma_j\}$, may be non-convex in $\theta$, if $\Lambda = (\mathbb{R}^+)^J$ then even finding a point where $\Phi$ is finite may be intractable. As such, we must make two standard concessions. First, we will assume $\Lambda$ is a compact convex set of bounded diameter. Intuitively, this bounds the penalty applied when the constraints are not satisfied. Further, instead of guaranteeing that Algorithm 1 finds an approximate minimizer of $\Phi$, we will show the algorithm finds an approximate stationary point. We note that stationarity is a standard convergence measure in non-convex optimization, and provide more discussion in Appendix D.1. Our subsequent analysis in fact provides for a slightly stronger, but more technical, notion of stationarity than we provide here; see Appendix D.6 for more details.

**Definition 5.1.** *($(\alpha, \nu)$-stationary point) A point $\theta$ is an $(\alpha, \nu)$-stationary point if $\exists \theta'$ s.t. $\|\theta - \theta'\| \leq \nu$ and $\min_{v \in \partial \Phi(\theta')} \|v\| \leq \alpha$, with $\partial \Phi$ the subdifferential of $\Phi$.*

When the loss is Lipschitz and smooth, SGDA converges to an approximate stationary point of $\Phi$ (Lin et al., 2020). Unfortunately, Algorithm 1 may have *biased* gradients, and the scale of noise present in $\hat{g}_\theta$ and $\hat{g}_\lambda$ may vary dramatically depending on $d$ and $J$. Thus, our main goal in this section is two-fold. First, we aim to formally show that despite using biased gradients, Algorithm 1 provably finds an approximate stationary point. Specifically, when the error in the primal updates (w.r.t. $\|\cdot\|_2$) is at most $\tau_\theta$ and the error in the dual updates is at most $\tau_\lambda$ (w.r.t. $\|\cdot\|_\infty$), we show SGDA on a nonconvex-linear loss finds a stationary point roughly with $\alpha = O(\frac{\sqrt{1+\tau_\theta}}{T^{1/4}} + \tau_\theta + \sqrt{\tau_\lambda} + \frac{1}{\sqrt{T}})$; see Appendix D.6. Second, we characterize the impact of the noise in $\hat{g}_\theta$ and $\hat{g}_\lambda$ added due to privacy. This involves correctly balancing the number of iterations $T$ with the scale of noise needed to ensure privacy, which increases with $T$. This leads to the following result for Algorithm 1 run without clipping.

**Theorem 5.2** (Informal). *Let $n = \min_{q \in [Q]} \{|D_q|\}$. Assume $h(\theta; x)$ and $\ell(\theta)$ are both Lipschitz and smooth. Then, under appropriate choices of parameters, Algorithm 1 is $(\epsilon, \delta)$-DP and with probability at least $1 - \rho$ there exists $t \in [T]$ s.t. $\theta_t$ is an $(\alpha, \alpha)$-stationary point of $\Phi$ with,*

$$\alpha = O\left( \left( \frac{\sqrt{d\log(\frac{JKn}{\rho})}\log(\frac{n}{\delta})}{n\epsilon} \right)^{\frac{1}{3}} + \frac{K^{\frac{1}{4}}\sqrt{\log(\frac{n}{\delta})}\log^{\frac{1}{4}}(\frac{JKn}{\rho})}{(n\epsilon)^{1/4}} \right),$$

*up to dependence on problem constants. We provide a complete statement and full proof in Appendix D.2. In Appendix D.7, we derive Lipschitz and smoothness constants for $\mathcal{L}$ from those of the classifier.*

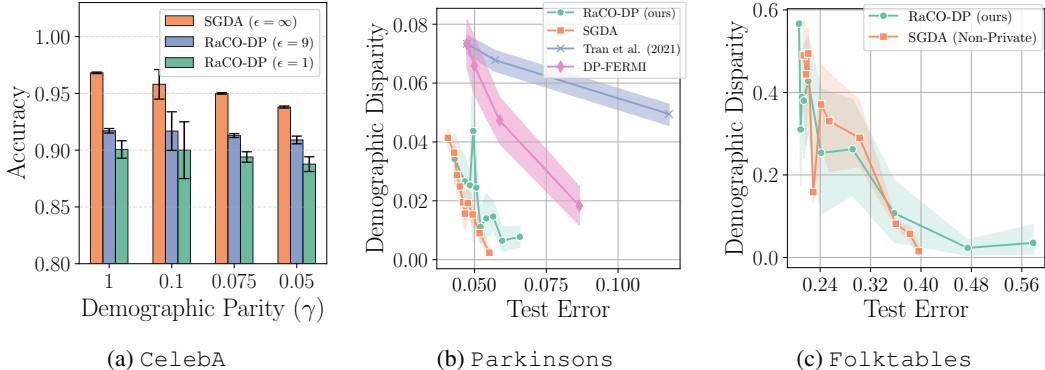

Figure 2: Fairness–utility trade-off for RaCO-DP over three benchmarks under demographic parity constraints training: (a) ResNet16 (5 runs), (b) and (c) logistic regression with $\varepsilon = 1$ (20 runs)

There are several key steps involved in achieving this result. First, we provide a general convergence proof for SGDA under the assumption that gradients have bounded error (Theorem D.5). Notably, in contrast to previous analyses, this result 1) allows for biased gradients; 2) depends on the $\ell_\infty$ error in the dual gradient estimate (rather than the $\ell_2$ error); and 3) achieves faster convergence (in terms of $T$) by leveraging the linear structure of the dual. To point (3), our analysis shows SGDA in this setting can converge as fast as $\frac{1}{T^{1/4}}$ instead of $\frac{1}{T^{1/6}}$ (shown by Lin et al., 2020), essentially matching the rate observed in comparable minimization (rather than min-max) settings. See Theorem D.5 for this specific result. The next step in proving Theorem 5.2 is to control the error of the gradient estimates while balancing the noise necessary for privacy. We show that this error scales proportional to $\tilde{O}\left(\frac{\sqrt{dT}}{n\epsilon} + \frac{1}{\sqrt{n}}\right)$ in the primal and $\tilde{O}\left(\frac{\sqrt{KT}\log J}{n\epsilon} + \sqrt{\frac{\log J}{n\epsilon}}\right)$ in the dual. We defer the reader to Lemma D.4 in Appendix D for a more detailed accounting of this error.

## 6 EXPERIMENTS

We empirically demonstrate that RaCO-DP: (i) achieves SOTA performance on standard tabular benchmarks; (ii) scales for deep neural networks, maintaining high utility under strong privacy guarantees; (iii) scales to a large number of constraints, and (iv) is computationally efficient.

**Setup.** We evaluate RaCO-DP on three constraint types: *demographic parity* (on Adult (Becker & Kohavi, 1996), Credit-Card (Yeh, 2009), Parkinsons Little (2007), ACSEmployment (Ding et al., 2021), CelebA (Liu et al., 2015)), *false negative rate* (on Adult, Heart Alex Teboul), and *equalized odds* (on Credit-Card). We benchmark against DP-FERMI (Lowy et al., 2023), Tran et al. (2021), and Jagielski et al. (2019), reporting their results from Lowy et al. (2023) where available. In settings without prior DP work (e.g., FNR constraints, deep networks on CelebA), we use non-private SGDA as a reference. All DP methods use $\delta = 10^{-5}$, with privacy loss tracked via the numerical accountant of Doroshenko et al. (2022). Full implementation and hyperparameter details are in Appendix E.2. For constraint definitions, see Appendix A.

**Tabular data.** On standard tabular benchmarks, Figure 2b (together with Figure 4, Figure 5, and Figure 6 in Appendix E.3) shows that across multiple datasets (Adult, Credit-Card, Parkinsons) and fairness constraints (demographic parity and equalized odds), RaCO-DP trains logistic regression models that consistently achieve higher accuracy at any fixed fairness disparity compared to all baselines under the same privacy budgets. Additional ablations for small $\varepsilon$ values are provided in Appendix E.8.

**Neural networks.** To demonstrate that RaCO-DP is applicable to large, non-convex models, we train a ResNet16 (He et al., 2016) (6.4M parameters) on CelebA under demographic parity constraints. As shown in Figure 2a, RaCO-DP achieves 90% accuracy with only a 10% demographic disparity gap for $\varepsilon = 1$. This result is close to the non-private model's 95% accuracy, showing that RaCO-DP is a practical tool for implementing fairness constraints in deep learning pipelines without a prohibitive loss in utility. Hyperparameter and training details are in Appendix E.7.

**Scaling.** To evaluate performance under a large constraint set, we train a logistic regression on `ACSEmployment`, enforcing demographic parity across 18 groups simultaneously. Figure 2c shows our method achieves a competitive utility–fairness trade-off even under strong privacy ($\varepsilon = 1$), confirming its effectiveness.

**Matching non-private rates.** We observe that RaCO-DP nearly matches non-private accuracy sometimes, especially for logistic regression on tasks such as `Adult`, `Credit-Card`, and `Parkinsons` (see Appendix E.3). In these cases, the sample size is large relative to the model's dimensionality, and convergence is fast (often within a few dozen updates). This allows training with large batches that reduce noise and still benefit from privacy amplification by subsampling. It also keeps the noise multiplier $\sigma$ (and Laplace parameter $b$) small, since the privacy budget is spread across only a handful of steps. These favorable conditions (large datasets, low-dimensional models, and rapid convergence) explain why the privacy gap narrows, consistent with prior observations (Song et al., 2013a). Once we employ higher-capacity models like ResNet16 on `CelebA` (see Figure 2a), the gap widens.

**Satisfiability.** The results in Figure 3 demonstrate that RaCO-DP consistently achieves the pre-specified constraint values $\gamma$ as measured by the *hard* rate constraints. This marks a significant improvement over existing approaches, which typically rely on indirect hyperparameter tuning to influence constraint satisfaction. Our direct Lagrangian formulation offers practitioners a simpler and more reliable method for constraint optimization. Additionally, these results show that tempered-sigmoid temperature $\tau = 1$ is sufficient to enforce hard constraints in practice.

**Computational efficiency.** We benchmark the average time per step for SGD, DP-SGD, RaCO-DP, and DP-FERMI via their public code (Gupta, 2023). On identical hardware, RaCO-DP trains *three orders of magnitude* faster than DP-FERMI on `Adult` (Appendix E.5).

**Limitations.** Koloskova et al. (2023) shows that gradient clipping biases SGD, which in our setting can push convergence outside the feasible set, making the clipping norm $C$ a critical hyperparameter. To illustrate this general issue (not specific to RaCO-DP), we impose a strict FNR $= 0$ constraint in logistic regression on `Adult` without DP noise ($\sigma = 0$, $b = \infty$). As shown in Figure 8a of Appendix E, norms below 12.5 prevent RaCO-DP from meeting the constraint, though weaker constraints allow smaller norms. Another limitation is the use of soft constraints for

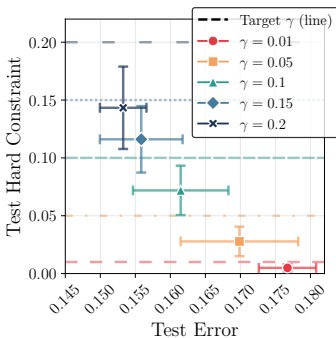

Figure 3: Constraint satisfaction on `Adult`.

the dual update. Hard constraints can be applied in practice, although this departs from the theoretical guarantees of RaCO-DP; as shown in Appendix E.4, this modification offers limited utility benefits.

## 7 RELATED WORK

To the best of our knowledge, private learning under general rate constraints has not been explored in prior work, except in the specific case of group fairness constraints. Accordingly, we review related work at the intersection of differential privacy and fairness, a key application area where early research has identified fundamental trade-offs between these two objectives (Cummings et al., 2019).

Existing work can be grouped into three main categories. A first line of research (Bagdasaryan et al.; Farrand et al., 2020; Suriyakumar et al., 2021; Tran et al., 2021; Kulynych et al., 2022; Esipova et al., 2022; Tran & Fioretto, 2023; Mangold et al., 2023) examines how privacy mechanisms can inadvertently harm fairness. For instance, Esipova et al. (2022) characterizes the disparate impact of DP-SGD due to gradient misalignment caused by clipping. The second line of work focuses on protecting the privacy of sensitive attributes used to enforce fairness constraints (Jagielski et al., 2019; Mozannar et al., 2020; Tran et al., 2023). Unlike standard DP, which protects the entire dataset, privacy for sensitive attributes requires injecting less noise, leading to improved model performance. However, this approach has significant limitations. Individuals can still be re-identified through their non-sensitive attributes, and if sensitive and non-sensitive features are correlated, an adversary may still be able to infer sensitive attributes. Due to these vulnerabilities, such works are not directly comparable to ours.

The third line of work, closest to ours, seeks to jointly enforce DP and group fairness (Berrada et al., 2023; Lowy et al., 2023). Berrada et al. (2023) use DP-SGD without fairness mitigation,

finding well-generalized models show no major privacy-fairness trade-off. However, their notion of fairness is error disparity, arguably measuring subpopulation generalization. Our work shows that while mitigation is needed for demographic parity, appropriate algorithm design can surmount the fairness-privacy trade-off. Lowy et al. (2023) propose a proxy objective for stochastic optimization of group fairness measures. In contrast, RaCO-DP supports a broad range of rate constraints that can be freely combined, without needing a task-specific objective. While its theoretical convergence rate is slower, RaCO-DP offers greater generality, complicating direct comparisons. Notably, Lowy et al. (2023) presents results for both sensitive attribute DP and standard DP (Definition 2.1), but their public code and evaluations focus on the weaker notion. Despite a stronger privacy guarantee, RaCO-DP achieves superior privacy-fairness trade-offs. We note that in contrast to Lowy et al. (2023), which seeks to satisfy the fairness objective by running an optimization procedure on a proxy objective, our optimization algorithm more directly targets the rate constraint of interest.

Fairness aside, SGDA is well-studied for minimax optimization (Nemirovski et al., 2009; Heusel et al.; Lin et al., 2020), with Yang et al. (2022) proposing its first DP analogue. Private minmax optimization has been heavily studied recently (Boob & Guzmán, 2023; Zhang et al., 2022; Bassily et al., 2023; 2024; Gonzalez et al., 2024), though work on non-convex losses is limited to (Lowy et al., 2023).

## 8 CONCLUSION

We introduced a DP algorithm using private histograms for training rate-constrained models. RaCO-DP demonstrates strong performance across various datasets and constraint types, often nearing non-private baselines while meeting privacy and constraint criteria. Our findings suggest that privacy-fairness trade-offs may be less significant than previously believed. Future work could explore private learning under individual fairness constraints, which cannot be formulated as rate constraints.

## 9 ACKNOWLEDGMENTS

We would like to acknowledge our sponsors, who support our research with financial and in-kind contributions: Apple, CIFAR through the Canada CIFAR AI Chair, Meta, Microsoft, NSERC through the Discovery Grant and an Alliance Grant with ServiceNow and DRDC, the Ontario Early Researcher Award, the Schmidt Sciences foundation through the AI2050 Early Career Fellow program. Resources used in preparing this research were provided, in part, by the Province of Ontario, the Government of Canada through CIFAR, and companies sponsoring the Vector Institute. Michael Menart is also supported by the Natural Sciences and Engineering Research Council of Canada (NSERC), grant RGPIN-2021-03206. The work of Tudor Cebere and Aurélien Bellet is supported by grant ANR-20-CE23-0015 (Project PRIDE) and the ANR 22-PECY-0002 IPOP (Interdisciplinary Project on Privacy) project of the Cybersecurity PEPR. Tudor Cebere is also sponsored by a Google Fellowship. This work was performed using HPC resources from GENCI–IDRIS (Grant 2023-AD011014018R1).

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

# A  APPLICATION TO OTHER RATE CONSTRAINTS

## A.1  FAIRNESS CONSTRAINTS

Fairness in machine learning aims to prevent models from making biased decisions based on sensitive attributes. We aim to train a classifier under fairness constraints by formulating a constrained optimization problem. We consider two popular group fairness Barocas et al. (2023) metrics: demographic parity and equality of odds. All group fairness measures can be formulated as rate-constraints Cotter et al. (2019b), for individual fairness (Dwork et al., 2012) it is easier to bound the per-sample contribution and privatize it with clipping and noising in the style of DP-SGD, thus a rate-constrained solution is not required, hence we focus on group fairness metrics.

**Definition A.1** (Demographic Parity). *A classifier $h(\theta; \cdot)$ satisfies demographic parity with respect to sensitive attribute $Z \in \mathcal{Z} = \{1, ..., |\mathcal{Z}|\}$ if the probability of predicting any class $k$ is independent of $Z$:*

$$\Pr[\hat{Y} = k \mid Z = z] = \Pr[\hat{Y} = k], \quad \forall z \in \mathcal{Z}, \forall k \in \mathcal{Y},$$

*where $\hat{Y} = h(\theta; x)$ is the predicted label.*

In practice, we do not have access to the true probabilities, so it is common to estimate them by empirical prediction rates $P_k$. Using a slack parameter $\gamma$, this gives:

$$P_k(D[Z = z]; \theta) - P_k(D[Z \neq z]; \theta) \leq \gamma \quad \forall z \in \mathcal{Z}, \forall k \in \mathcal{Y} \tag{13}$$

Demographic parity thus leads to $J = |\mathcal{Z}| \cdot |\mathcal{Y}|$ rate constraints of the form specified in Eqn. (6). The global partition is of size $Q = |\mathcal{Z}|$ with elements $D_z = D[Z = z], \forall z \in \mathcal{Z}$, and for the constraint corresponding to elements $z \in \mathcal{Z}$ and $y \in \mathcal{Y}$, we have $\mathcal{I} = \{\{z\}, [|\mathcal{Z}|] \setminus z\}$. The associated vector $\alpha$ has $\alpha_{\{z\}, y} = 1$ and $\alpha_{\{[|\mathcal{Z}|] \setminus z\}, y} = -1$, with the rest of the components set to 0.

**Definition A.2** (Equality of Odds). *A classifier $h(\theta; \cdot)$ satisfies equality of odds if the probability of predicting any class $k$ is conditionally independent of the sensitive attribute $Z$ given the ground truth:*

$$\Pr[\hat{Y} = k \mid Y = k', Z = z] = \Pr[Y = k', \hat{Y} = k, Z = z'], \quad \forall z', z \in \mathcal{Z}, \forall k, k' \in \mathcal{Y} \tag{14}$$

We note that the original notion of equalized odds is for binary sensitive attribute. For non-binary sensitive attributes, we can extend equalized odds to equalize rates between all subpopulations (as above), or we can consider a counter-factual definition of equalized odds:

$$\Pr[\hat{Y} = k \mid Y = k', Z = z] = \Pr[Y = k', \hat{Y} = k, Z \neq z], \quad \forall, z \in \mathcal{Z}, \forall k, k' \in \mathcal{Y} \tag{15}$$

In the above formulation, we seek to achieve equal odds for each subpopulation compared to other subpopulations combined (e.g. white vs. non-white, etc.). It is clear that in the binary sensitive attribute, the definitions are the same. Our framework can handle either variant by changing the adjusting the local partitioning (see Section 3) but we adopt the counter-factual definition.

We observe that the only difference between the equality of odds and demographic parity is the additional conditioning on the ground truth, which we will reflect as the additional predicate $Y = k'$ in our base rates to define the following constraint:

$$P_k(D[Y = k', Z = z]; \theta) - P_k(D[Y = k', Z = z']; \theta) \leq \gamma \quad \forall z \in \mathcal{Z}, \forall k, k' \in \mathcal{Y} \tag{16}$$

Equality of odds leads to $J = |\mathcal{Y}|^2 \times |\mathcal{Z}|$ number of constraints. With regards to implementing Eqn. (6), we can use a global partition with $|\mathcal{Y}| \times |\mathcal{Z}|$ where each element is the subset of $D$ with some fixed ground truth label $k$ and class $z$. The constraint for some $k \in [K]$ and $z \in \mathcal{Z}$ then has $\mathcal{I}$ which specifies the local partition $\{D[Y = k, Z = z], D[Y = k, Z \neq z]\}$ with the corresponding vector $\alpha$ having a $+1$ coefficient corresponding to a prediction rate of $D[Y = k', Z = z]$.

Many other group fairness constraints exist but they are all reducible to base rate constraints in a similar manner. Note the similarity between Equations (7) and (16), where the only difference is the additional conditioning on ground truth labels $Y$ in equality of odds.

| Objective | Formula | Number of Constraints |
|---|---|---|
| Demographic Parity | $P_k(D[Z = z]; \theta) - P_k(D[Z \neq z]; \theta) \leq \gamma$ | $\forall k \in \mathcal{Y}$ (predicted), $\forall z \in \mathcal{Z}$ (sens.) |
| Equality of Odds | $P_k(D[Y = k', Z = z]; \theta) - P_k(D[Y = k', Z \neq z]; \theta) \leq \gamma$ | $\forall k, k' \in \mathcal{Y}$ (predicted and g.t.) $\forall z \in \mathcal{Z}$ (sens. attr.) |
| False Negative Rate | $P_k(D[Y \neq k]; \theta) \leq \gamma$ | $\forall k \in \mathcal{Y}$ (predicted) |

Table 1: **Rate Constraints.** Given a dataset $D$, $C_k(D)$ is the prediction counts for class $k$, and $P_k(D) = C_k(D)/|D|$ is the prediction rate. $D[\text{pred}]$ indicates the subset of $D$ where predicate $\text{pred}$ is true, e.g., $D[Y = y, Z = z]$ is the subset of $D$ with sensitive attribute (sen. attr.) $Z = z$ and ground truth (g.t.) labels $Y = y$.

## A.2   FALSE NEGATIVE RATE

**Definition A.3.** *(False Negative Rate (FNR)) A classifier's false negative rate (FNR) measures how often it incorrectly predicts negative for samples that are actually positive. More formally, a classifier satisfies a false negative rate constraint if*

$$P_k(D[Y \neq k]; \theta) \leq \gamma \quad \text{for } k \in [|\mathcal{Y}|] \tag{17}$$

Assuming the constraint is well-defined, FNR leads $J = |\mathcal{Y}|$ rate constraints of the form in Eqn. (6), with the global partition of size $Q = |Y|$ with elements $D_y = D[Y = y], y \in \mathcal{Y}$. For the constraint corresponding to a fixed $y \in \mathcal{Y}$, we have $\mathcal{I} = \{\mathcal{Y}/y\}$ with an associated $\alpha_{\{\mathcal{Y}/y\}, y} = 1$.

# B  PRIVACY ANALYSIS

**Theorem B.1.** *Let $\sigma \geq 10 \max \left\{ \frac{C \log(T/\delta)}{r|D|\epsilon}, \frac{C\sqrt{T} \log(T/\delta)}{|D|\epsilon} \right\}$ and $b \geq 2 \max \left\{ \frac{1}{\epsilon}, \frac{r\sqrt{T \log(T/\delta)}}{\epsilon} \right\}$, then Algorithm 1 is $(\epsilon, \delta)$-DP.*

*Proof.* The $\ell_2$-sensitivity of $\left( \sum_{x \in B^{(t)}} \mathrm{clip}(g_{x,\theta}^{(t)}, \frac{C}{r|D|}) \right)$ is clearly at most $\frac{C}{r|D|}$. Thus the standard guarantees of the Gaussian mechanism ensures $(\frac{1}{2}\epsilon_1, \frac{1}{2}\delta_1)$-DP w.r.t. the minibatch, where $\epsilon_1 \leq \min\{1, \frac{1}{r\sqrt{8T \log(1/\delta)}}\}$ and $\delta_1 = \frac{\delta}{2T}$. Similarly, because $\{D_1, ..., D_Q\}$ is a partition, the $\ell_1$-sensitivity of the histogram is at most 1, and so the guarantees of the Laplace mechanism ensure $(\frac{1}{2}\epsilon_1, 0)$-DP w.r.t. to the minibatch. By composition, the combined mechanism is $(\epsilon_1, \delta_1)$-DP w.r.t. the minibatch. Since this mechanism acts a Poisson subsampled portion of the dataset and $\epsilon' \leq 1$, the privacy w.r.t. the overall dataset is $(\epsilon_2, \frac{1}{2}\delta_2)$ with $\epsilon_2 = r\epsilon_1 \leq \frac{1}{\sqrt{8T \log(1/\delta)}}$ and $\delta_2 \leq \frac{r\delta}{2T}$. Now applying advanced composition, the overall privacy of Algorithm 1 over $T$ rounds is $(\epsilon_3, \delta_3)$-DP with $\epsilon_3 \leq \sqrt{8T \log(1/\delta)}\epsilon_2 \leq \epsilon$ and $\delta_3 \leq (T+1)\delta_2 \leq \delta$. □

# C  TECHNICAL LEMMAS

**Lemma C.1.** *Let $X$ and $Y$ be sums of $k_X$ and $k_Y$ zero-centered Laplace random variables with scale parameter $b$, respectively, and let $\mu_X, \mu_Y > 0$, $\frac{\mu_x}{\mu_y} \leq 1$, $k_X \leq k_Y$. For any $\rho \in (0, 1)$, if $\mu_Y \geq 4k_Y b \ln \left( \frac{1}{2\rho} \right)$ then it holds that,*

$$P \left[ \left| \frac{\mu_X + X}{\mu_Y + Y} - \frac{\mu_X}{\mu_Y} \right| < \frac{4k_Y b}{\mu_y} \ln \left( \frac{8}{\rho} \right) \right] \geq 1 - \rho. \tag{18}$$

*Proof.* We have,

$$P \left[ \left| \frac{\mu_X + X}{\mu_Y + Y} - \frac{\mu_X}{\mu_Y} \right| < \epsilon \right]$$

$$\geq P \left[ \left| \frac{X}{\mu_Y + Y} \right| + \left| \frac{\mu_X Y}{\mu_Y(\mu_Y + Y)} \right| < \epsilon, \mu_Y + Y > \frac{\mu_Y}{2} \right] \quad \text{(triangle inequality)}$$

$$\geq P \left[ \left| \frac{2X}{\mu_Y} \right| + \left| \frac{2\mu_X Y}{\mu_Y^2} \right| < \epsilon, \mu_Y + Y > \frac{\mu_Y}{2} \right] \quad \text{(conditioning on } \mu_Y + Y > \frac{\mu_Y}{2} \text{)}$$

$$\geq P \left[ \left| \frac{2X}{\mu_Y} \right| \leq \frac{\epsilon}{2}, \left| \frac{2Y}{\mu_Y} \right| < \frac{\epsilon}{2}, \mu_Y + Y > \frac{\mu_Y}{2} \right] \quad \text{(using } \mu_x \leq \mu_y \text{)}$$

$$\geq 1 - P \left[ |X| \geq \frac{\mu_Y \epsilon}{4} \right] - P \left[ |Y| \geq \frac{\mu_Y \epsilon}{4} \right] - P \left[ Y \leq -\frac{\mu_Y}{2} \right] \quad \text{(Negation \& Union Bound)}$$

$$\geq 1 - 2\exp(-\frac{\mu_Y \epsilon}{4k_Y b}) - 2\exp(-\frac{\mu_Y \epsilon}{4k_X b}) - \frac{1}{2}\exp(-\frac{\mu_Y}{k_Y b}) \quad \text{(concentration for Laplace R.Vs \& Laplace CDF)}$$

$$\geq 1 - 2\exp(-\frac{\mu_Y \epsilon}{4k_Y b}) - 2\exp(-\frac{\mu_Y \epsilon}{4k_X b}) - \frac{1}{2}\exp(-\frac{\mu_Y}{k_Y b}) \quad (k_Y \geq k_X)$$

$$\geq 1 - 4\exp(-\frac{\mu_Y \epsilon}{4k_Y b}) - \frac{1}{2}\exp(-\frac{\mu_Y}{k_Y b})$$

$$\geq 1 - 4\exp(-\frac{\mu_Y \epsilon}{4k_Y b}) - \frac{\rho}{2}$$

the last inequality uses that $\mu_Y \geq 4k_Y b \ln \left( \frac{1}{2\rho} \right)$. Now setting $\epsilon = \frac{4k_Y b}{\mu_y} \ln \left( \frac{8}{\rho} \right)$ yeilds,

$$P \left[ \left| \frac{\mu_X + X}{\mu_Y + Y} - \frac{\mu_X}{\mu_Y} \right| < \frac{4k_Y b \ln(8/\rho)}{\mu_y} \right] \geq 1 - \rho. \tag{19}$$

□

**Lemma C.2** (Error of sampled rates). *Let $p = \frac{\sum_{x_i \in X} x_i}{|X|}$ with $x_i \in [0,1]$ and $p_r = \frac{\sum_{x_i \in X_q} x_i}{r|X|}$ where $X_r$ is obtained by performing Poisson sampling on $X$ with probability $r$. If $|X|\, r \geq \log(1/\rho)$ then (up to an order):*

$$P\left[|p - p_r| \leq \sqrt{\frac{\log(1/\rho)}{r|X|}}\right] \geq 1 - \rho \tag{20}$$

*Proof.* We have,

$$
\begin{aligned}
&P\left[|p - p_r| \leq \epsilon\right] \\
&= P\left[\left|\frac{\sum_{X_i \in X} X_i}{|X|} - \frac{\sum_{X_i \in X} \text{Bern}(r) X_i}{r|X|}\right| \leq \epsilon\right] && (X_r \sim \text{Poisson}(X, r)) \\
&= P\left[\left|r \sum_{X_i \in X} X_i - \sum_{X_i \in X} \text{Bern}(r) X_i\right| \leq \epsilon r |X|\right] \\
&\geq 1 - \exp\left(-\frac{\epsilon^2 r^2 |X|^2}{2\left(\text{Var}(\sum_{X_i \in X} \text{Bern}(r) X_i) + \epsilon r |X|/3\right)}\right) && (\text{Bernstein Ineq.}) \\
&\geq 1 - \exp\left(-\frac{\epsilon^2 r^2 |X|}{2\left(r(1-r) + \epsilon r/3\right)}\right) && (X_i \in [0,1]; \text{Var}(\text{Bern}(r)) = r(1-r)) \\
&\geq 1 - \exp\left(-\frac{\epsilon^2 r |X|}{2\left(1/4 + \epsilon/3\right)}\right) && r(1-r) \leq 1/4 \\
\implies &\epsilon \geq 2\max\left\{\frac{\log(1/\rho)}{r|X|}, \sqrt{\frac{\log(1/\rho)}{r|X|}}\right\} \\
&\geq 2\sqrt{\frac{\log(1/\rho)}{r|X|}} && (\text{using the assumption that } |X|\, r \geq \log(1/\rho))
\end{aligned}
$$

$\square$

# D    MISSING DETAILS FROM SECTION 5

## D.1    ADDITIONAL BACKGROUND ON STATIONARY POINT DEFINITION

For a smooth function $f : \theta \mapsto \mathbb{R}$, a standard notion of (first order) stationarity would involve bounding the norm of the gradient. However, for non-smooth functions, this notion does not accurately capture convergence. For example, if $f(\theta) = \|\theta\|$, a point may be arbitrarily close to the minimum, but still have gradient norm 1. To address this discrepancy, alternative notions of stationarity for non-smooth functions have been introduced, such as Definition 5.1. In the example where $f(\theta) = \|\theta\|$, this relaxation allows points which are *close* to the cusp at $\theta = \mathbf{0}$, whereas a bound on the gradient norm would allow *only* the point $\theta = \mathbf{0}$ for any non-trivial bound on the gradient. In fact, our convergence proof yields a slightly stronger notion of stationarity known as proximal near stationarity Davis & Drusvyatskiy (2019). We elect to present Definition 5.1 as it requires less background information.

## D.2    PROOF OF THEOREM 5.2

In this section we will detail the proof of Theorem 5.2 and provide a more precise theorem statement. Before doing so, we will introduce some important notation.

For any $I \subseteq [Q]$, let $D_I = \underset{i \in I}{\cup} D_i$. Given a subset $B \subset D$, define $B_{\cap I} = B \cap (\underset{i \in I}{\cup} D_i)$ and recall that $\mathcal{I}_j$ and $\alpha_j \in \mathbb{R}^{|\mathcal{I}_j| \times K}$ denote the corresponding family of susbets of $[Q]$ and weight vector associated with constraint $\Gamma_j$. Let $n = \min_{q \in [Q]} \{|D_q|\}$.

Let $\|\Lambda\|_1$ be the $\ell_1$ diameter of $\Lambda$. Let $G_\ell$ and $G_h$ be the $\ell_2$ Lipschitz constants w.r.t. $\theta$ of $h$ and $\ell$ respectively. Similarly, let $\beta_\ell$ and $\beta_h$ be the corresponding $\ell_2$-smoothness constants. We recall

the temperature parameter of the softmax is denoted as $\tau$. Let $c_1 = \max_{j \in [J]} \|\alpha_j\|$. Note that many rate constraint only compare two prediction rates, and so $c_1$ is typically at most 2. Define $\hat{\Phi}(\theta) = \min_{\theta'}\{\Phi(\theta') + \beta\|\theta - \theta'\|^2\}$ and $\hat{\Phi}_0 = \hat{\Phi}(\theta_0) - \min_\theta\{\hat{\Phi}(\theta)\}$ and $\mathcal{L}_0 = \mathcal{L}(\theta_0, \lambda_0) - \min_{\lambda,\theta}\{\mathcal{L}(\lambda,\theta)\}$ (see Section D.6 for more details on these quantities). We can now present the more complete version of convergence result.

**Theorem D.1.** *Assume* $n \geq \frac{1}{r}\max\{\ln\left(\frac{2c_1 JT}{\rho}\right), 8\frac{Kb}{r}\ln\left(\frac{2T}{\rho}\right), 8\log(J|D|TK/\rho\}$. *Then, under appropriate choices of parameters, Algorithm 1 run without clipping is $(\epsilon, \delta)$-DP and with probability at least $1 - \rho$ there exists $t \in [T]$ s.t. $\theta_t$ is an $(\alpha, \alpha/[2\beta])$-stationary point of $\Phi$ with*

$$\alpha = O\Bigg(\left(\left(\hat{\Phi}_0\beta G^2\right)^{1/4} + \sqrt{\beta\mathcal{L}_0} + G\right)\Bigg(\Big(\frac{\sqrt{d}\log(n/\delta)\sqrt{\log(JKn/\rho)}}{n\epsilon}\Big)^{1/3}$$

$$+ \frac{K^{1/4}\sqrt{\beta\|\Lambda\|_1}\sqrt{\log(n/\delta)}(\log(JKn/\rho))^{1/4}}{(n\epsilon)^{1/4}}\Bigg)\Bigg),$$

*where $G = G_\ell + c\tau G_h\|\Lambda\|_1$ and $\beta = \beta_\ell + 2c\tau \cdot \max\left\{G_h\sqrt{J}, \|\Lambda\|_1(2G_h + \tau\beta_h)\right\}$.*

Proving this statement will involve several major steps. First, in Section D.4 we derive the necessary noise levels needed to ensure that Algorithm 1 is private. Second, in Section D.5 we bound the error in the gradients at each time step. Next, in Section D.6 we give a general convergence rate for SGDA under the condition that the gradients have bounded error. Finally, we derive the overall Lipschitz and smoothness constants of $\mathcal{L}$ based on the base smoothness and Lipschitz constants in Section D.7. These results are then combined in Section D.3 to obtain the final result.

In one final remark, we note the following fact will be used in several places.

**Lemma D.2.** *Let $n \geq 4\log(J|D|/\rho)$ and $t \in [T]$. With probability at least $1 - \rho$ it holds for every $j \in [J]$ and $I \in \mathcal{I}_j$ that $|B_{\cap I}^{(t)}| \geq \frac{1}{2}r|I|n$.*

*Proof.* By Lemma C.2 we have for any $j \in [J]$ and $I \in \mathcal{I}_j$,

$$P\left[r|D_I| - |B_{\cap I}^{(t)}| \geq \sqrt{r|D_I|\log(1/\gamma)}\right] \leq \gamma.$$

Thus since $|D_I| \geq n \geq \log(J|D|/\rho)$, it holds with probability at least $1 - \rho$, for every $j \in [J]$ and $I \in \mathcal{I}_j$ that $|B_{\cap I}| \geq r|D_I| - \sqrt{r|D_I|\log(1/\rho)} \geq 0.5r|D_I| \geq 0.5r|I|n$. $\qquad\square$

### D.3 PROOF OF THEOREM D.1

With the previously established results, we can now verify a setting of parameters which proves the theorem statement. Specifically, we set,

$$T = \min\left\{\left(\frac{n\epsilon}{\sqrt{d}}\right)^{4/3}, \frac{n\epsilon}{K}\right\}, \qquad \sigma = \frac{G\sqrt{T}\log(T/\delta)}{n\epsilon}, \qquad b = \frac{r\sqrt{T\log(T/\delta)}}{\epsilon}.$$

Note that by Theorem D.3, this ensures that Algorithm 1 is $(\epsilon, \delta)$-DP so long as $r \geq \frac{1}{\sqrt{T}}$.

Using Lemma D.4 can now instantiate Theorem D.5. For $\tau_\theta$ we have,

$$\tau_\lambda = O\left(\frac{c_1 Kb\log(JKn/\rho)}{rn} + c_1\sqrt{\frac{\log\left(\frac{JKn}{\rho}\right)}{rn}}\right)$$

$$= O\left(\frac{c_1 K\sqrt{T}\log(T/\delta)\log(JKn/\rho)}{n\epsilon} + c_1\sqrt{\frac{\log\left(\frac{JKn}{\rho}\right)}{rn}}\right).$$

Setting $\tau_\lambda$ as the quantity above, we can write the bound on $\tau_\theta$ as,

$$\tau_\theta = O\left(\sqrt{d}\sigma\sqrt{\log(4T/\rho)} + \frac{4G_\ell\sqrt{\log(4/\rho)}}{\sqrt{r|D|}} + \frac{4G_\ell\sqrt{\log(4/\rho)}}{\sqrt{r|D|}} + \|\Lambda\|_1\tau G_h\tau_\theta\right)$$

$$= O\left(G\left(\frac{\sqrt{dT\log(T/\rho)}\log(T/\delta)}{n\epsilon} + \frac{\sqrt{\log(1/\rho)}}{\sqrt{r|D|}}\right) + \|\Lambda\|_1\tau G_h\tau_\theta\right).$$

Now, in the non-trivial regime where $\tau_\theta \leq G$, Theorem D.5 implies that for $r$ large enough,

$$\alpha = O\left(\frac{\left(\hat{\Phi}_0\beta(G^2 + \tau_\theta^2)\right)^{1/4}}{T^{1/4}} + \tau_\theta + \sqrt{\beta\|\Lambda\|_1\tau_\lambda} + \frac{\sqrt{\beta\mathcal{L}_0}}{\sqrt{T}}\right)$$

$$= O\left(\left(\left(\hat{\Phi}_0\beta G^2\right)^{1/4} + \sqrt{\beta\mathcal{L}_0} + G\right)\left(\left(\frac{\sqrt{d}\log(T/\delta)\sqrt{\log(JKn/\rho)}}{n\epsilon}\right)^{1/3}\right.\right.$$

$$\left.\left.+ \frac{K^{1/4}\sqrt{\beta\|\Lambda\|_1}\sqrt{\log(T/\delta)}(\log(JKn/\rho))^{1/4}}{(n\epsilon)^{1/4}}\right)\right).$$

## D.4 PRIVACY OF ALGORITHM 1 UNDER LIPSCHITZNESS

**Theorem D.3.** *Assume $h$ and $\ell$ are $G_\ell$ and $G_h$ Lipschitz. Then for some universal constant $c$ and $\sigma \geq c(c_1\|\Lambda\|_1\tau G_h + G_\ell)\max\left\{\frac{\log(T/\delta)}{rn\epsilon}, \frac{C\sqrt{T}\log(T/\delta)}{n\epsilon}\right\}$ and $b \geq 2\max\left\{\frac{1}{\epsilon}, \frac{r\sqrt{T\log(T/\delta)}}{\epsilon}\right\}$, then Algorithm 1 is $(\epsilon, 3\delta)$-DP.*

*Proof.* First, by Lemma D.2 and the conditions of Theorem D.1, probability at least $1 - \delta$, for every $t \in [T]$, $j \in [J]$ and $I \in [I]$ it holds that $|B_{\cap I}| \geq 0.5r|I| \cdot n$. Consequently, the concentration of Laplace noise and the conditions of Theorem D.1 imply $\sum_{i \in I}\sum_{k \in [K]} \hat{H}_{i,k}^{(t)} \geq 0.25rn$ with probability at least $1 - 2\delta$. Conditional on this event, the $\ell_2$-sensitivity of $\left(\sum_{x \in B^{(t)}} g_{x,\theta}^{(t)}\right)$ is at most $\frac{G_\ell}{r|D|} + \frac{c_1\tau G_h}{0.25rn}$ since then,

$$\|\nabla\hat{R}(\theta, \lambda; H, x)\| \leq \sum_{j=1}^{J}\sum_{I \in \mathcal{I}_j}\sum_{k \in K}\frac{\lambda_j\alpha_{j,I,k}}{0.25rn}\|\nabla_\theta[\sigma_\tau(h(\theta;x))_k]\| \leq \frac{c_1\|\Lambda\|\tau G_h}{0.25rn}.$$

Thus, the scale of Gaussian noise implies that the releasing the primal gradient is $(\frac{1}{2}\epsilon_1, \frac{1}{2}\delta_1)$-DP w.r.t. the minibatch, where $\epsilon_1 \leq \min\{1, \frac{1}{r\sqrt{T\log(1/\delta)}}\}$ and $\delta_1 = \frac{\delta}{T}$. From here one can follow the same steps as in the proof of Theorem 4.1 to obtain an overall privacy of $(\epsilon, \delta)$-DP conditional on the previously mentioned event that each $B_{\cap I}$ is large. Since this event happens with probability at least $1 - 2\delta$, we obtain a final overall privacy guarantee of $(\epsilon, 3\delta)$-DP. $\qquad\square$

## D.5 BOUNDING GRADIENT ERROR

**Lemma D.4.** *Let $\rho \in [0, 1]$ and $t \in [T]$. Under the assumptions of Theorem D.1, conditional on $\theta^{(t)}, \lambda^{(t)}$, it holds with probability at least $1 - 2\rho$ that,*

$$\|g_\theta^{(t)} - \nabla_\theta\mathcal{L}(\theta^{(t)}, \lambda^{(t)})\|_2 \leq \sigma\sqrt{d\log(4/\rho)} + \frac{4G_\ell\sqrt{\log(4/\rho)}}{\sqrt{r|D|}}$$

$$+ \|\Lambda\|_1\tau G_h\left(\frac{8c_1Kb\log(64JKn/\rho)}{rn} + c_1\sqrt{\frac{\log\left(\frac{8JKn}{\rho}\right)}{rn}}\right),$$

$$\|g_\lambda^{(t)} - \nabla_\lambda\mathcal{L}(\theta^{(t)}, \lambda^{(t)})\|_\infty \leq \frac{8c_1Kb\log(64JKn/\rho)}{rn} + c_1\sqrt{\frac{\log\left(\frac{8JKn}{\rho}\right)}{rn}}.$$

*Proof.* We will bound each error term separately.

**Error of the Dual Gradient.**

We start with the following bound,

$$P\left[\|\hat{g}_\lambda^{(t)} - \nabla_\lambda \mathcal{L}(\theta^{(t)}, \lambda^{(t)})\|_\infty \geq \epsilon\right] = P\left[\max_j \left\{\left|\sum_{I \in \mathcal{J}_j}\sum_{k=1}^K \alpha_{j,I,k} P_k(D_I) - \sum_{I \in \mathcal{J}_j}\sum_{k_1 \in K} \frac{\alpha_{j,I,k_1}\sum_{i \in I}\hat{H}_{i,k}^{(t)}}{\sum_{i \in I}\sum_{k_2 \in K}\hat{H}_{i,k_2}^{(t)}}\right|\right\} \geq \epsilon\right]$$

$$\leq \sum_j^J \sum_{I \in \mathcal{J}_j}\sum_{k=1}^K \mathbb{1}\left[|\alpha_{j,I,k} \neq 0|\right] P\left[\left|P_k(D_I) - \frac{\sum_{i \in I}\hat{H}_{i,k_1}^{(t)}}{\sum_{i \in I}\sum_{k_2 \in K}\hat{H}_{i,k_2}^{(t)}}\right| \geq \frac{\epsilon}{c_1}\right].$$

(21)

We thus have for any $\epsilon_1 + \epsilon_2 = \epsilon$ that,

$$P\left[\left|P_k(D_I) - \frac{\sum_{i \in I}\hat{H}_{i,k_1}^{(t)}}{\sum_{i \in I}\sum_{k_2 \in K}\hat{H}_{i,k_2}^{(t)}}\right| \geq \frac{\epsilon}{c_1}\right] \leq P\left[|P_k(D_I) + P_k(B_{\cap I})| + \left|P_k(B_{\cap I}) - \frac{\sum_{i \in I}\hat{H}_{i,k_1}^{(t)}}{\sum_{i \in I}\sum_{k_2 \in K}\hat{H}_{i,k_2}^{(t)}}\right| \geq \frac{\epsilon}{c_1}\right]$$

$$\leq 1 - P\left[|P_k(D_I) + P_k(B_{\cap I})| + \left|P_k(B_{\cap I}) - \frac{\sum_{i \in I}\hat{H}_{i,k_1}^{(t)}}{\sum_{i \in I}\sum_{k_2 \in K}\hat{H}_{i,k_2}^{(t)}}\right| \leq \frac{\epsilon}{c_1}\right]$$

$$\leq 1 - P\left[|P_k(D_I) + P_k(B_{\cap I})| \leq \frac{\epsilon_1}{c_1}, \left|P_k(B_{\cap I}) - \frac{\sum_{i \in I}\hat{H}_{i,k_1}^{(t)}}{\sum_{i \in I}\sum_{k_2 \in K}\hat{H}_{i,k_2}^{(t)}}\right| \leq \frac{\epsilon_2}{c_1}\right]$$

$$\leq P\left[|P_k(D_I) + P_k(B_{\cap I})| \geq \frac{\epsilon_1}{c_1}\right] + P\left[\left|P_k(B_{\cap I}) - \frac{\sum_{i \in I}\hat{H}_{i,k_1}^{(t)}}{\sum_{i \in I}\sum_{k_2 \in K}\hat{H}_{i,k_2}^{(t)}}\right| \geq \frac{\epsilon_2}{c_1}\right].$$

We will start by bounding $P\left[\left|P_k(B_{\cap I}) - \frac{\sum_{i \in I}\hat{H}_{i,k}^{(t)}}{\sum_{i \in I}\sum_{k_2 \in K}\hat{H}_{i,k_2}^{(t)}}\right| \geq \frac{\epsilon_1}{c_1}\right]$ for any fixed $I$ and $k$. Observe that

conditional on $|B_{\cap I}|$, the sampling process is equivalent to drawing $|B_{\cap I}|$ samples uniformly at random from $|D_I|$ without replacement. Therefore by Lemma C.1 we have,

$$P\left[\left|P_k(B_{\cap I}) - \frac{\sum_{i \in I}\hat{H}_{i,k}^{(t)}}{\sum_{i \in I}\sum_{k_2 \in K}\hat{H}_{i,k_2}^{(t)}}\right| \geq \frac{4K|I|b\log(16JKn/\rho)}{|B_{\cap I}|}\bigg||B_{\cap I}|\right] \leq \frac{\rho}{4JKn}.$$

Now by Lemma D.2, $P\left[|B_{\cap}I| \leq \frac{r}{2}|I|n\right] \leq \frac{\rho}{4JKn}$, and so,

$$P\left[\left|P_k(B_{\cap I}) - \frac{\sum_{i \in I}\hat{H}_{i,k}^{(t)}}{\sum_{i \in I}\sum_{k_2 \in K}\hat{H}_{i,k_2}^{(t)}}\right| \geq \frac{8Kb\log(16JKn/\rho)}{rn}\right] \leq \frac{\rho}{2JKn}.$$

Thus it suffices to have $\epsilon_1 = \frac{8c_1 b\log(16JKn/\rho)}{rn}$.

Looking now at the statistical error term and applying Lemma C.2 we obtain:

$$P\left[|P_k(B_{\cap I}) - P_k(D_I)| \geq \sqrt{\frac{\log\left(\frac{2JKn}{\rho}\right)}{r|D_I|}}\right] \leq \frac{\rho}{2JKn}.$$

(22)

Observing that $\sqrt{\frac{\log\left(\frac{2JKn}{\rho}\right)}{r|D_I|}} \leq \sqrt{\frac{\log\left(\frac{2JKn}{\rho}\right)}{rn}}$, one can see it suffices for $\epsilon_2 = c_1\sqrt{\frac{\log\left(\frac{2JKn}{\rho}\right)}{rn}}$.

Plugging $\epsilon = \epsilon_1 + \epsilon_2$ back into Eqn. (21) we obtain

$$P\left[\|\hat{g}_\lambda^{(t)} - \nabla_\lambda \mathcal{L}(\theta^{(t)}, \lambda^{(t)})\|_\infty \geq \frac{8c_1 b \log(16JKn/\rho)}{rn} + c_1 \sqrt{\frac{\log\left(\frac{2JKn}{\rho}\right)}{rn}}\right]$$

$$\leq \sum_j^J \sum_{I \in \mathcal{J}_j} \sum_{k=1}^K \mathbb{1}\left[|\alpha_{j,I,k} \neq 0|\right] \frac{\rho}{JKn} \leq \rho.$$

This proves the claim.

**Error in Primal Gradient.** First observe that

$$\|\hat{g}_\theta - \nabla_\theta \mathcal{L}(\theta^{(t)}, \lambda^{(t)})\| \leq \|Z^{(t)}\| + \left\|\nabla_\theta \ell(\theta^{(t)}) - \frac{1}{r|D|} \sum_{x \in B^{(t)}} \nabla_\theta \ell(\theta^{(t)}; x)\right\|$$

$$+ \left\|\nabla_\theta R(\theta^{(t)}, \lambda^{(t)}) - \sum_{x \in B^{(t)}} \hat{R}(\theta^{(t)}, \lambda^{(t)}; \hat{H}, x)\right\|$$

For $\hat{g}_\theta \in \mathbb{R}^d$, using the concentration of Gaussian noise we obtain,

$$P[\|Z^{(t)}\| \geq \sigma \sqrt{d \log(4/\rho)}] \leq \frac{\rho}{4}.$$

For the second term, by Bernstein's inequality we have

$$P\left[\left\|\left(\sum_{i \in B^{(t)}} \nabla_\theta \ell(\theta, \lambda; x_i)\right) - r|D|\nabla_\theta \ell(\theta_t, \lambda_t)\right\| \geq \alpha\right] \leq \exp\left(-\frac{\alpha^2/2}{\alpha G_\ell/3 + |D|G_\ell^2 \max\{r, 1/2\}}\right).$$

Thus with probability at least $1 - \frac{\rho}{4}$ one has that,

$$\left\|\left(\frac{1}{r|D|} \sum_{i \in B^{(t)}} \nabla_\theta \ell(\theta, \lambda, x_i)\right) - \nabla_\theta \ell(\theta_t, \lambda_t)\right\| \leq 4 \max\left\{\frac{G_\ell \sqrt{\log(4/\rho)}}{\sqrt{r|D|}}, \frac{G_\ell \log(4/\rho)}{r|D|}\right\}.$$

And so if $r|D| \geq \log(4/\rho)$ we have with probability at least $1 - \rho/4$ that,

$$\left\|\nabla_\theta \ell(\theta^{(t)}) - \frac{1}{r|D|} \sum_{x \in B^{(t)}} \nabla_\theta \ell(\theta^{(t)}; x)\right\| \leq \frac{4G_\ell \sqrt{\log(4/\rho)}}{\sqrt{r|D|}}.$$

For the regularizer we have,

$$\left\|\nabla_\theta R(\theta^{(t)}, \lambda^{(t)}) - \sum_{x \in B^{(t)}} \hat{R}(\theta^{(t)}, \lambda^{(t)}; \hat{H}, x)\right\|$$

$$\leq \left\|\sum_{x \in D} \sum_{j=1}^J \sum_{I \in \mathcal{I}_j} \sum_{k_1 \in [K]} \frac{\lambda_j \, \alpha_{j,I,k_1} \, \mathbb{1}_{[x \in D_I]}}{\sum_{i \in I} \sum_{k_2 \in [K]} H_{i,k_2}^{(t)}} \nabla_\theta[\sigma_\tau(h(\theta; x)_{k_1}] - \sum_{x \in D} \sum_{j=1}^J \sum_{I \in \mathcal{I}_j} \sum_{k_1 \in [K]} \frac{\lambda_j \, \alpha_{j,I,k_1} \, \mathbb{1}_{[x \in B_{\cap I}]}}{\sum_{i \in I} \sum_{k_2 \in [K]} \hat{H}_{i,k_2}^{(t)}} \nabla_\theta[\sigma_\tau(h(\theta; x)_{k_1}]\right\|$$

$$\leq \left|\sum_{x \in D} \sum_{j=1}^J \sum_{I \in \mathcal{I}_j} \sum_{k_1 \in [K]} \frac{\lambda_j \, \alpha_{j,I,k_1} \, \mathbb{1}_{[x \in D_I]}}{\sum_{i \in I} \sum_{k_2 \in [K]} H_{i,k_2}^{(t)}} - \sum_{x \in D} \sum_{j=1}^J \sum_{I \in \mathcal{I}_j} \sum_{k_1 \in [K]} \frac{\lambda_j \, \alpha_{j,I,k_1} \, \mathbb{1}_{[x \in B_{\cap I}]}}{\sum_{i \in I} \sum_{k_2 \in [K]} \hat{H}_{i,k_2}^{(t)}}\right| \tau G_h$$

$$\leq \left|\sum_{j=1}^J \lambda_j \sum_{I \in \mathcal{I}_j} \sum_{k_1 \in [K]} \alpha_{j,I,k_1} \left(\frac{|D_I|}{\sum_{i \in I} \sum_{k_2 \in [K]} H_{i,k_2}^{(t)}} - \frac{|B_{\cap I}|}{\sum_{i \in I} \sum_{k_2 \in [K]} \hat{H}_{i,k_2}^{(t)}}\right)\right| \tau G_h$$

$$\leq \left(\sum_{j=1}^J \lambda_j \left|\sum_{I \in \mathcal{I}_j} \sum_{k_1 \in [K]} \frac{\alpha_{j,I,k_1} |D_I|}{\sum_{i \in I} \sum_{k_2 \in [K]} H_{i,k_2}^{(t)}} - \sum_{I \in \mathcal{I}_j} \sum_{k_1 \in [K]} \frac{\alpha_{j,I,k_1} |B_{\cap I}|}{\sum_{i \in I} \sum_{k_2 \in [K]} \hat{H}_{i,k_2}^{(t)}}\right|\right) \tau G_h$$

The term inside the absolute value can be bounded using the same analysis used in bounding the dual gradient error. Thus we have with probability at least $1 - \frac{\rho}{4}$ that,

$$\left\| \nabla_\theta R(\theta^{(t)}, \lambda^{(t)}) - \sum_{x \in B^{(t)}} \hat{R}(\theta^{(t)}, \lambda^{(t)}; \hat{H}, x) \right\| \leq \|\Lambda\|_1 \tau G_h \left( \frac{8c_1 K b \log(64JKn/\rho)}{rn} + c_1 \sqrt{\frac{\log\left(\frac{8JKn}{\rho}\right)}{rn}} \right).$$

Combining the above bounds yields the claimed bound on the primal gradient error.

$\square$

## D.6 CONVERGENCE OF SGDA

The overall structure of our convergence proof is similar to that of Lin et al. (2020), but with several significant modifications. Most significantly, our proof explicitly leverages the linear structure of the dual to improve the convergence rate for our application. This linear structure also allows our analysis to depend on an $\| \cdot \|_\infty$ bound on the gradient error when $\Lambda$ has bounded $\| \cdot \|_1$ diameter. This in contrast to previously existing analysis which depend on the $\| \cdot \|_2$ error of the dual gradient, which could be much worse in our case due to the noise added for privacy. Separately, our analysis also differs from Lin et al. (2020) in that it accounts for potential bias in the gradient estimates and tracks the disparate impact the scale of noise in $g_\theta$ and $g_\lambda$ may have on the convergence rate.

In order to present our proof, we start with some necessary preliminaries. Let $\Phi(\theta) = \max_{\lambda \in \Lambda} \{\mathcal{L}(\theta, \lambda)\}$. Let $\hat{\Phi}$ denote the Moreau envelope of $\Phi$ with parameter $2\beta$. That is, $\hat{\Phi}(\theta) = \min_{\theta'} \{\Phi(\theta') + \beta\|\theta - \theta'\|^2\}$. Let $\Delta^{(t)} = \Phi(\theta^{(t)}) - \mathcal{L}(\theta^{(t)}, \lambda^{(t)})$ for all $t \in \{0, ..., T\}$. Further, we define $\lambda^*(\theta) = \arg\max_{\lambda \in \Lambda} \{\mathcal{L}(\theta, \lambda)\}$. We denote the proximal operator of a function $f$ as $\text{prox}_f(\theta) = \arg\min_\theta \{f(\theta') + \frac{1}{2}\|\theta - \theta'\|^2\}$. It is known that under the condition that $f$ is $\beta$-smooth and $\Lambda$ is bounded, that $\hat{\Phi}$ is differentiable with $\nabla\hat{\Phi}(\theta) = 2\beta(\theta - \text{prox}_{\Phi/[2\beta]}(\theta))$, and that any point $\theta$ for which which $\|\nabla\hat{\Phi}(\theta)\| \leq \alpha$ is an $(\alpha, \alpha/[2/\beta])$-stationary point with respect to Definition 5.1; see Lin et al. (2020, Lemma 3.8). Also, under these conditions, $\Phi$ is $G$-Lipschitz. We defer the reader towards Lin et al. (2020) for more details on these statements.

We present the following statement, which gives a convergence rate for Algorithm 1 in terms of the amount of noise added.

**Theorem D.5.** *Define $\hat{\Phi}_0 = \hat{\Phi}(\theta_0) - \min_\theta \left\{\hat{\Phi}(\theta)\right\}$ and $\mathcal{L}_0 = \mathcal{L}(\theta_0, \lambda_0) - \min_{\lambda, \theta} \{\mathcal{L}(\lambda, \theta)\}$. Assume $\mathcal{L}(\cdot, \cdot)$ is $\beta$-smooth, $\mathcal{L}(\cdot, \lambda)$ is $G$-Lipschitz for all $\lambda \in \Lambda$, and $\mathcal{L}(\theta, \cdot)$ is linear for all $\theta \in \mathbb{R}^d$. Conditional on the event that for all $t \in \{0, ..., T-1\}$, $\|g_{\theta,t} - \nabla_\theta\mathcal{L}(\theta^{(t)}, \lambda^{(t)})\|_2 \leq \tau_\theta$ and $\|g_\lambda^{(t)} - \nabla_\lambda\mathcal{L}(\theta^{(t)}, \lambda^{(t)})\|_\infty \leq \tau_\lambda$, when Algorithm 1 is run with $\eta_\lambda \geq \left(\frac{\eta_\theta G(G+\tau_\theta)}{\tau_\lambda}\right)$ and $\eta_\theta = \sqrt{\frac{\hat{\Phi}_0}{2T\beta(G^2+\tau_\theta^2)}}$ there exists $t \in \{0, ..., T-1\}$ such that $\theta_t$ is an $(\alpha, \alpha/[2\beta])$-stationary point with*

$$\alpha = O\left( \frac{\left(\hat{\Phi}_0\beta(G^2 + \tau_\theta^2)\right)^{1/4}}{T^{1/4}} + \tau_\theta + \sqrt{\beta\|\Lambda\|_1\tau_\lambda} + \frac{\sqrt{\beta\mathcal{L}_0}}{\sqrt{T}} \right).$$

We will prove this statement by showing that Algorithm 1 finds a point where the gradient of $\hat{\Phi}$ is small. Note this is sufficient as Lin et al. (2020, Lemma 3.8) implies that a point, $\theta$, for which $\|\nabla\hat{\Phi}(\theta)\| \leq \alpha$ is an $(\alpha, \alpha/[2\beta])$-stationary point with respect to Definition 5.1.

We will break the majority of the proof into three distinct lemmas. The first lemma gives a bound on the decrease in $\hat{\Phi}$.

**Lemma D.6.** *Under the assumptions of Theorem D.5, the iterates of Algorithm 1 satisfy for any $t \in [T-1]$,*

$$\hat{\Phi}(\theta^{(t)}) - \hat{\Phi}(\theta^{(t-1)}) \leq -\frac{\eta_\theta}{4}\|\nabla\hat{\Phi}(\theta^{(t-1)})\|^2 + 2\beta\eta_\theta\Delta^{(t-1)} + 2\beta\eta_\theta^2(G^2 + \tau_\theta^2) + 2\eta_\theta\tau_\theta^2. \quad (23)$$

*Proof.* Let $\hat{\theta}^{(t-1)} = \text{prox}_{\hat{\Phi}}(\theta^{(t-1)})$. By the definition of the Moreau envelope we have

$$\hat{\Phi}(\theta^{(t)}) \leq \Phi(\hat{\theta}^{(t-1)}) + \beta\|\hat{\theta}^{(t-1)} + \theta^{(t)}\|^2. \tag{24}$$

Using the update rule we have

$$
\begin{aligned}
\|\hat{\theta}^{(t-1)} - \theta^{(t)}\|^2 &= \|\hat{\theta}^{(t-1)} - \theta^{(t-1)} + \eta_\theta g_\theta^{(t-1)}\|^2 \\
&= \|\hat{\theta}^{(t-1)} - \theta^{(t-1)}\|^2 + 2\eta_\theta \left\langle \hat{\theta}^{(t-1)} - \theta^{(t-1)}, g_\theta^{(t-1)} \right\rangle + \eta_\theta^2 \|g_\theta^{(t-1)}\|^2 \\
&\leq \|\hat{\theta}^{(t-1)} - \theta^{(t-1)}\|^2 + 2\eta_\theta \left\langle \hat{\theta}^{(t-1)} - \theta^{(t-1)}, \nabla_\theta \mathcal{L}(\theta^{(t-1)}, \lambda^{(t-1)}) \right\rangle \\
&\quad + 2\eta_\theta \left\langle \hat{\theta}^{(t-1)} - \theta^{(t-1)}, g_\theta^{(t-1)} - \nabla_\theta \mathcal{L}(\theta^{(t-1)}, \lambda^{(t-1)}) \right\rangle + 2\eta_\theta^2 (G^2 + \tau_\theta^2)
\end{aligned}
$$

Plugging this back into Eqn. (24) and using the definition of the Moreau envelope we obtain,

$$
\begin{aligned}
\hat{\Phi}(\theta^{(t)}) &\leq \hat{\Phi}(\hat{\theta}^{(t-1)}) + 2\beta\eta_\theta \left\langle \hat{\theta}^{(t-1)} - \theta^{(t-1)}, \nabla_\theta \mathcal{L}(\theta^{(t-1)}, \lambda^{(t-1)}) \right\rangle \\
&\quad + 2\beta\eta_\theta \left\langle \hat{\theta}^{(t-1)} - \theta^{(t-1)}, g_\theta^{(t-1)} - \nabla_\theta \mathcal{L}(\theta^{(t-1)}, \lambda^{(t-1)}) \right\rangle + 2\beta\eta_\theta^2 (G^2 + \tau_\theta^2) \tag{25}
\end{aligned}
$$

By way of bounding the third term on the RHS above, we use Young's inequality to derive,

$$2\beta\eta_\theta \left\langle \hat{\theta}^{(t-1)} - \theta^{(t-1)}, g_\theta^{(t-1)} - \nabla_\theta \mathcal{L}(\theta^{(t-1)}, \lambda^{(t-1)}) \right\rangle \leq \frac{\beta^2 \eta_\theta}{2}\|\hat{\theta}^{(t-1)} - \theta^{(t-1)}\|^2 + 2\eta_\theta \|g_\theta^{(t-1)} - \nabla_\theta \mathcal{L}(\theta^{(t-1)}, \lambda^{(t-1)})\|^2.$$

Plugging this into the above, we now have the following derivation,

$$
\begin{aligned}
&\hat{\Phi}(\theta^{(t)}) - \hat{\Phi}(\hat{\theta}^{(t-1)}) \\
&\leq 2\beta\eta_\theta \left\langle \hat{\theta}^{(t-1)} - \theta^{(t-1)}, \nabla_\theta \mathcal{L}(\theta^{(t-1)}, \lambda^{(t-1)}) \right\rangle + \frac{\beta^2 \eta_\theta}{2}\|\hat{\theta}^{(t-1)} - \theta^{(t-1)}\|^2 + 2\beta\eta_\theta^2(G^2 + \tau_\theta^2) + 2\eta_\theta \tau_\theta^2 \\
&\overset{(i)}{\leq} 2\beta\eta_\theta \left( \mathcal{L}(\hat{\theta}^{(t-1)}, \lambda^{(t-1)}) - \mathcal{L}(\theta^{(t-1)}, \lambda^{(t-1)}) + \frac{3\beta}{4}\|\hat{\theta}^{(t-1)} - \theta^{(t-1)}\|^2 \right) + 2\beta\eta_\theta^2(G^2 + \tau_\theta^2) + 2\eta_\theta \tau_\theta^2 \\
&\leq 2\beta\eta_\theta \left( \Phi(\hat{\theta}^{(t-1)}) - \mathcal{L}(\theta^{(t-1)}, \lambda^{(t-1)}) + \frac{3\beta}{4}\|\hat{\theta}^{(t-1)} - \theta^{(t-1)}\|^2 \right) + 2\beta\eta_\theta^2(G^2 + \tau_\theta^2) + 2\eta_\theta \tau_\theta^2 \\
&= 2\beta\eta_\theta \left( \Phi(\hat{\theta}^{(t-1)}) + \Phi(\theta^{(t-1)}) - \Phi(\theta^{(t-1)}) - \mathcal{L}(\theta^{(t-1)}, \lambda^{(t-1)}) + \frac{3\beta}{4}\|\hat{\theta}^{(t-1)} - \theta^{(t-1)}\|^2 \right) \\
&\quad + 2\beta\eta_\theta^2(G^2 + \tau_\theta^2) + 2\eta_\theta \tau_\theta^2 \\
&\overset{(ii)}{\leq} 2\beta\eta_\theta \left( -\beta\|\hat{\theta}^{(t-1)} - \theta^{(t-1)}\|^2 + \Delta^{(t-1)} + \frac{3\beta}{4}\|\hat{\theta}^{(t-1)} - \theta^{(t-1)}\|^2 \right) + 2\beta\eta_\theta^2(G^2 + \tau_\theta^2) + 2\eta_\theta \tau_\theta^2 \\
&= -\frac{\eta_\theta \beta^2}{2}\|\hat{\theta}^{(t-1)} - \theta^{(t-1)}\|^2 + 2\beta\eta_\theta \Delta^{(t-1)} + 2\beta\eta_\theta^2(G^2 + \tau_\theta^2) + 2\eta_\theta \tau_\theta^2 \\
&\overset{(iii)}{=} -\frac{\eta_\theta}{2}\|\nabla\hat{\Phi}((\theta^{(t-1)})\|^2 + 2\beta\eta_\theta \Delta^{(t-1)} + 2\beta\eta_\theta^2(G^2 + \tau_\theta^2) + 2\eta_\theta \tau_\theta^2
\end{aligned}
$$

Above, $(i)$ uses the fact that $\hat{\theta}^{(t-1)}$ is generated by the proximal operator. Inequality $(ii)$ uses the fact the definitions of the Moreau envelope and $\Delta^{(t-1)}$, i.e. $\|\hat{\theta}^{(t-1)} - \theta^{(t-1)}\|^2 = \frac{1}{4\beta^2}\|\nabla\hat{\Phi}(\theta^{(t-1)})\|^2$. Equality $(iii)$ uses properties of the Moreau envelope. $\square$

The next two lemmas pertain to bounding the $\Delta^{(t)}$ terms.

**Lemma D.7.** *Under the conditions of Theorem D.5, for any $t \in [T]$ and $s \leq t - 1$ one has,*

$$
\begin{aligned}
\Delta^{(t-1)} &\leq \eta_\theta G(G + \tau_\theta)(2t - 2s - 1) + \frac{1}{2\eta_\lambda}(\|\lambda^*(\theta^{(s)}) - \lambda^{(t-1)}\|^2 - \|\lambda^*(\theta^{(s)}) - \lambda^{(t)}\|^2) + 2\|\Lambda\|_1 \tau_\lambda \\
&\quad + [\mathcal{L}(\theta^{(t)}, \lambda^{(t)}) - \mathcal{L}(\theta^{(t-1)}, \lambda^{(t-1)})]. \tag{26}
\end{aligned}
$$

*Proof.* Let $s \le t - 1$. By adding and subtracting terms we have

$$
\begin{aligned}
\Delta^{(t-1)} &= [\mathcal{L}(\theta^{(t-1)}, \lambda^*(\theta^{(t-1)})) - \mathcal{L}(\theta^{(t-1)}, \lambda^*(\theta^{(s)}))] + [\mathcal{L}(\theta^{(t)}, \lambda^{(t)}) - \mathcal{L}(\theta^{(t-1)}, \lambda^{(t-1)})] \\
&\quad + [\mathcal{L}(\theta^{(t-1)}, \lambda^{(t)}) - \mathcal{L}(\theta^{(t)}, \lambda^{(t)})] + [\mathcal{L}(\theta^{(t-1)}, \lambda^*(\theta^{(s)})) - \mathcal{L}(\theta^{(t-1)}, \lambda^{(t)}))] \\
&\le [\mathcal{L}(\theta^{(t-1)}, \lambda^*(\theta^{(t-1)})) - \mathcal{L}(\theta^{(t-1)}, \lambda^*(\theta^{(t-1)}))] + [\mathcal{L}(\theta^{(t-1)}, \lambda^*(\theta^{(s)})) - \mathcal{L}(\theta^{(t-1)}, \lambda^*(\theta^{(s)}))] \\
&\quad + [\mathcal{L}(\theta^{(t)}, \lambda^{(t)}) - \mathcal{L}(\theta^{(t-1)}, \lambda^{(t-1)})] + [\mathcal{L}(\theta^{(t-1)}, \lambda^{(t)}) - \mathcal{L}(\theta^{(t)}, \lambda^{(t)})] + [\mathcal{L}(\theta^{(t-1)}, \lambda^*(\theta^{(s)})) - \mathcal{L}(\theta^{(t-1)}, \lambda^{(t)}))] \\
&\le G(G + \tau_\theta)[2\|\theta^{(t-1)} - \theta^{(s)}\| + \|\theta^{(t-1)} - \theta^{(t)}\|] + [\mathcal{L}(\theta^{(t)}, \lambda^{(t)}) - \mathcal{L}(\theta^{(t-1)}, \lambda^{(t-1)})] \\
&\quad + [\mathcal{L}(\theta^{(t-1)}, \lambda^*(\theta^{(s)})) - \mathcal{L}(\theta^{(t-1)}, \lambda^{(t)}))] \\
&\le \eta_\theta G(G + \tau_\theta)(2t - 2s - 1) + [\mathcal{L}(\theta^{(t)}, \lambda^{(t)}) - \mathcal{L}(\theta^{(t-1)}, \lambda^{(t-1)})] + [\mathcal{L}(\theta^{(t-1)}, \lambda^*(\theta^{(s)})) - \mathcal{L}(\theta^{(t-1)}, \lambda^{(t)}))].
\end{aligned}
$$

To complete the lemma, we will bound the loss difference $\mathcal{L}(\theta^{(t-1)}, \lambda^*(\theta^{(s)})) - \mathcal{L}(\theta^{(t-1)}, \lambda^{(t)})$. Since $\mathcal{L}(\theta^{(t-1)}, \cdot)$ is linear we have,

$$
\begin{aligned}
\mathcal{L}(\theta^{(t-1)}, \lambda^{(t-1)}) - \mathcal{L}(\theta^{(t-1)}, \lambda^{(t)}) &= \left\langle \lambda^{(t-1)} - \lambda^{(t)}, \nabla_\lambda \mathcal{L}(\theta^{(t-1)}, \lambda^{(t)}) \right\rangle \\
&\le \left\langle \lambda^{(t-1)} - \lambda^{(t)}, g_\lambda^{(t)} \right\rangle + \left\langle \lambda^{(t-1)} - \lambda^{(t)}, \nabla_\lambda \mathcal{L}(\theta^{(t-1)}, \lambda^{(t)}) - g_\lambda^{(t)}) \right\rangle \\
&\le \left\langle \lambda^{(t-1)} - \lambda^{(t)}, g_\lambda^{(t)} \right\rangle + \|\lambda^{(t-1)} - \lambda^{(t)}\|_1 \cdot \|\nabla_\lambda \mathcal{L}(\theta^{(t-1)}, \lambda^{(t)}) - g_\lambda^{(t)})\|_\infty \\
&\le \left\langle \lambda^{(t-1)} - \lambda^{(t)}, g_\lambda^{(t)} \right\rangle + \|\Lambda\|_1 \tau_\lambda
\end{aligned}
$$

Now a standard analysis using the fact that $\lambda^{(t-1)} + g_\lambda^{(t)}$ is projected orthogonaly onto $\Lambda$ we have

$$
0 \le \|\lambda^{(t)} - \lambda^{(t-1)}\|^2 \le \frac{1}{2\eta_\lambda}\|\lambda^*(\theta^{(t-1)}) - \lambda^{(t-1)}\|^2 - \frac{1}{2\eta_\lambda}\|\lambda^*(\theta^{(t-1)}) - \lambda^{(t)}\|^2 + \lambda \left\langle g_\lambda^{(t)}, \lambda^{(t)} - \lambda^* \right\rangle
$$

Now by plugging back into the above yields,

$$
\begin{aligned}
&\mathcal{L}(\theta^{(t-1)}, \lambda^{(t-1)}) - \mathcal{L}(\theta^{(t-1)}, \lambda^{(t)}) \\
&\le \left\langle \lambda^*(\theta^{(s)}) - \lambda^{(t)}, g_\lambda^{(t)} \right\rangle + \|\Lambda\|_1 \tau_\lambda + \frac{1}{2\eta_\lambda}\|\lambda^*(\theta^{(t-1)}) - \lambda^{(t-1)}\|^2 - \frac{1}{2\eta_\lambda}\|\lambda^*(\theta^{(t-1)}) - \lambda^{(t)}\|^2.
\end{aligned}
$$

Using concavity we obtain,

$$
\begin{aligned}
\mathcal{L}(\theta^{(t-1)}, \lambda^*(\theta^{(s)})) - \mathcal{L}(\theta^{(t-1)}, \lambda^{(t)}) &\le \left\langle \lambda^*(\theta^{(s)}) - \lambda^{(t)}, g_\lambda^{(t)} - \nabla_\lambda \mathcal{L}(\theta^{(t-1)}, \lambda^{(t)}) \right\rangle + \|\Lambda\|_1 \tau_\lambda \\
&\quad + \frac{1}{2\eta_\lambda}\|\lambda^*(\theta^{(t-1)}) - \lambda^{(t-1)}\|^2 - \frac{1}{2\eta_\lambda}\|\lambda^*(\theta^{(t-1)}) - \lambda^{(t)}\|^2 \\
&\le \frac{1}{2\eta_\lambda}\|\lambda^*(\theta^{(t-1)}) - \lambda^{(t-1)}\|^2 - \frac{1}{2\eta_\lambda}\|\lambda^*(\theta^{(t-1)}) - \lambda^{(t)}\|^2 + 2\|\Lambda\|_1 \tau_\lambda.
\end{aligned}
$$

Plugging this back into the starting inequality achieves the claimed bound.

$\square$

**Lemma D.8.** *Under the conditions of Theorem D.5 it holds that,*

$$
\frac{1}{T}\sum_{t=0}^{T-1} \Delta^{(t)} \le \|\Lambda\|_1 \sqrt{\frac{\eta_\theta G(G + \tau_\theta)}{\eta_\lambda}} + 2\|\Lambda\|_1 \tau_\lambda + \frac{\mathcal{L}(\theta_T, \lambda_T) - \mathcal{L}(\theta_0, \lambda_0)}{T}.
$$

*Proof.* For any $s \in [T]$ and $M \in [T]$ one has by analyzing the telescoping sum created from Eqn.(26) that

$$
\begin{aligned}
\sum_{t=s}^{s+M-1} \Delta^{(t)} &\le \eta_\theta G(G + \tau_\theta)M^2 + \frac{1}{2\eta_\lambda}(\|\lambda^{(s)} - \lambda^*(\theta^{(s)})\|^2 + \|\lambda^{(s+M)} - \lambda^*(\theta^{(s)})\|^2) + 2M\|\Lambda\|_1 \tau_\lambda \\
&\quad + [\mathcal{L}(\theta^{(s+M)}, \lambda^{(s+M)}) - \mathcal{L}(\theta^{(s)}, \lambda^{(s)})] \\
&\le \eta_\theta G(G + \tau_\theta)M^2 + \frac{1}{\eta_\lambda}\|\Lambda\|_2^2 + 2M\|\Lambda\|_1 \tau_\lambda + [\mathcal{L}(\theta^{(s+M)}, \lambda^{(s+M)}) - \mathcal{L}(\theta^{(s)}, \lambda^{(s)})] \\
&\le \eta_\theta G(G + \tau_\theta)M^2 + \frac{1}{\eta_\lambda}\|\Lambda\|_1^2 + 2M\|\Lambda\|_1 \tau_\lambda + [\mathcal{L}(\theta^{(s+M)}, \lambda^{(s+M)}) - \mathcal{L}(\theta^{(s)}, \lambda^{(s)})].
\end{aligned}
$$

By applying this inequality over disjoint "blocks" of iterates, of which there are at most $T/M$, we can use this to obtain,

$$\frac{1}{T}\sum_{t=0}^{T-1}\Delta^{(t)} \le \eta_\theta G(G+\tau_\theta)M + \frac{1}{M\eta_\lambda}\|\Lambda\|_1^2 + 2\|\Lambda\|_1\tau_\lambda + \frac{\mathcal{L}_0}{T}.$$

We can now set $M = \frac{\|\Lambda\|_1}{\sqrt{\eta_\theta\eta_\lambda G(G+\tau_\theta)}}$ to obtain the desired inequality. $\qquad\square$

We can now prove the main theorem statement.

*Proof of Theorem D.5.* Recall $\hat{\Phi}_0 = \hat{\Phi}(\theta_0) - \min_\theta\left\{\hat{\Phi}(\theta)\right\}$ and $\mathcal{L}_0 = \mathcal{L}(\theta_0, \lambda_0) - \min_{\lambda,\theta}\left\{\mathcal{L}(\lambda,\theta)\right\}$. Summing over Eqn. (23) obtains,

$$\hat{\Phi}(\theta_{T-1}) \le \hat{\Phi}(\theta_0) + 2\beta\eta_\theta\left(\sum_{t=0}^{T-1}\Delta^{(t)}\right) + 2T[\beta\eta_\theta^2(G^2 + \tau_\theta^2) + \eta_\theta\tau_\theta^2] - \frac{\eta_\theta}{4}\left(\sum_{t=0}^{T-1}\|\nabla\hat{\Phi}(\theta^{(t)})\|^2\right).$$

Which implies for any $M \in [T]$,

$$\frac{1}{T}\left(\sum_{t=0}^{T-1}\|\nabla\hat{\Phi}(\theta^{(t)})\|^2\right) \le \frac{4\hat{\Phi}_0}{\eta_\theta T} + \frac{8\beta}{T}\left(\sum_{t=0}^{T-1}\Delta^{(t)}\right) + 4\beta\eta_\theta(G^2 + \tau_\theta^2) + 4\tau_\theta^2$$

$$\overset{(i)}{\le} \frac{4\hat{\Phi}_0}{\eta_\theta T} + 8\beta\eta_\theta(G^2 + \tau_\theta^2) + 4\tau_\theta^2$$

$$+ 8\beta\|\Lambda\|_1\sqrt{\frac{\eta_\theta G(G+\tau_\theta)}{\eta_\lambda}} + 8\beta\|\Lambda\|_1\tau_\lambda + \frac{8\beta\mathcal{L}_0}{T}.$$

Inequality $(i)$ above uses Eqn. (26). Setting $\eta_\lambda \ge \left(\frac{\eta_\theta G(G+\tau_\theta)}{\tau_\lambda}\right)$ yields,

$$\frac{1}{T}\left(\sum_{t=0}^{T-1}\|\nabla\hat{\Phi}(\theta^{(t)})\|^2\right) \le \frac{4\hat{\Phi}_0}{\eta_\theta T} + 8\beta\eta_\theta(G^2 + \tau_\theta^2) + 4\tau_\theta^2 + 16\beta\|\Lambda\|_1\tau_\lambda + \frac{8\beta\mathcal{L}_0}{T}.$$

Setting $\eta_\theta = \sqrt{\frac{\hat{\Phi}_0}{2T\beta(G^2+\tau_\theta^2)}}$ yields,

$$\frac{1}{T}\left(\sum_{t=0}^{T-1}\|\nabla\hat{\Phi}(\theta^{(t)})\|^2\right) \le \frac{16\sqrt{\hat{\Phi}_0\beta(G^2 + \tau_\theta^2)}}{\sqrt{T}} + 4\tau_\theta^2 + 16\beta\|\Lambda\|_1\tau_\lambda + \frac{16\beta\mathcal{L}_0}{T}.$$

Finally, this implies the claimed convergence.

$$\frac{1}{T}\left(\sum_{t=0}^{T-1}\|\nabla\hat{\Phi}(\theta^{(t)})\|\right) = O\left(\frac{\left(\hat{\Phi}_0\beta(G^2 + \tau_\theta^2)\right)^{1/4}}{T^{1/4}} + \tau_\theta + \sqrt{\beta\|\Lambda\|_1\tau_\lambda} + \frac{\sqrt{\beta\mathcal{L}_0}}{\sqrt{T}}\right).$$

$\qquad\square$

## D.7 REGULARITY PROPERTIES OF LAGRANGIAN

We will use the following standard fact about composing Lipschitz and/or smooth functions.

**Lemma D.9.** *Let $h : \mathbb{R}^d \mapsto \mathbb{R}^k$ be $G_h$-Lipschitz and $\beta_h$-smooth and $g : \mathbb{R}^k \mapsto \mathbb{R}$ be $G_g$-Lipschitz and $\beta_g$-smooth. Then $g \circ h$ is $(G_h G_g)$-Lipschitz and $(G_h\beta_g + G_g^2\beta_h)$-smooth.*

*Proof.* Let $\mathbf{J}_h(\theta)$ denote the Jacobian of $h$ at $\theta$. Since $h$ is $G_h$ Lipschitz, the spectral norm of the Jacobian is at most $G_h$, $\|\mathbf{J}_h(\theta)\|_2 \leq G_h$. Observe $\nabla_\theta g(h(\theta)) = \nabla g(h(\theta))^\top \mathbf{J}_h(\theta)$. Thus $\|\nabla_\theta g(h(\theta))\| \leq \|\nabla g(h(\theta))\| \cdot \|\mathbf{J}_h(\theta)\|_2 \leq G_g G_h$.

For the second part of the claim, observe that,

$$
\begin{aligned}
\|\nabla_\theta g(h(\theta)) - \nabla_\theta g(h(\theta'))\| &\leq \|\nabla g(h(\theta))\mathbf{J}_h(\theta) - \nabla g(h(\theta'))\mathbf{J}_h(\theta')\| \\
&= \|[\nabla g(h(\theta)) - \nabla g(h(\theta'))]\mathbf{J}_h(\theta') + \nabla g(h(\theta))[\mathbf{J}_h(\theta) - \mathbf{J}_h(\theta')]\| \\
&\leq (G_h^2 \beta_g + G_g \beta_h)\|\theta - \theta'\|.
\end{aligned}
$$

$\square$

In the following, we assume the predictor $h$ is $G_h$-Lipschitz and $\beta_h$-smooth with respect to $h$, and similarly for $\ell$ with parameters $G_\ell$ and $\beta_\ell$. Our aim is to derive regularity parameters for the Lagrangian given these base parameters. Note that the function which outputs one coordinate the tempered soft max is $\tau$-Lipschitz and $\tau$-smooth.

**Lemma D.10.** *Let $\Lambda \subset (\mathbb{R}^+)^{K \times Q}$ be a bounded set of diameter at most $\|\Lambda\|_1$ w.r.t. $\|\cdot\|_1$. Assume for any $j \in [J]$ that the vector $\alpha$ associated with rate constraint $j$ satisfies $\|\alpha\|_1 \leq c_1$ for some constant c. Then $\mathcal{L}(\cdot, \lambda)$ is G-Lipschitz with $G = G_\ell + c\tau G_h \|\Lambda\|_1$ for any $\lambda \in \Lambda$ and $\mathcal{L}$ is $\beta$-smooth with $\beta = \beta_\ell + 2c\tau \cdot \max\left\{G_h \sqrt{J}, \|\Lambda\|_1(2G_h + \tau\beta_h)\right\}$.*

Before presenting the proof, we note that $G_\ell$ and $\beta_\ell$ could also be further decomposed using the Lipschitz/smoothness constants of $\ell$ and $h$ via Lemma D.9. However, as these parameters are not affected by our approach in the way the regularity parameters of the regularizer are, we omit these more specific details.

*Proof.* Let for $I \subseteq [Q]$ let $D_I = \cup_{q \in I} D_q$. To establish Lipschitzness w.r.t. $\theta$, we have for any $\theta, \lambda$ that

$$
\begin{aligned}
\|\nabla_\theta \mathcal{L}(\theta, \lambda)\| &\leq \|\nabla_\theta \ell(\theta)\| + \sum_{j=1}^J \lambda_j \sum_{I \in \mathcal{J}_j} \sum_k^K \alpha_{j,I,k} \left\|\nabla_\theta P_k(D_I; \theta)\right\|_2 \\
&\leq G_\ell + \|\lambda\|_1 c_1 \cdot \max_{S \subseteq D, k \in [K]} \{\|\nabla_\theta P_k(S; \theta)\|\}
\end{aligned}
$$

Note that for any $S \subseteq D$ and $k \in [K]$ that $\|\nabla_\theta P_k(S; \theta, \tau)\| = \|\frac{1}{|S|}\sum_{x \in S}\nabla_\theta[\sigma_\tau(h(\theta; x))_k]\| \leq \tau G_h$. Plugging this into the above achieved the claimed Lipschitz parameter.

To prove $\mathcal{L}$ is smooth, we have for any $\theta, \theta' \in \mathbb{R}^d$ and $\lambda, \lambda' \in \Lambda$,

$$
\begin{aligned}
\|\nabla\mathcal{L}(\theta, \lambda) - \nabla\mathcal{L}(\theta', \lambda')\|^2 &\leq 2\|\nabla\mathcal{L}(\theta, \lambda) - \nabla\mathcal{L}(\theta, \lambda')\|^2 + 2\|\nabla\mathcal{L}(\theta, \lambda') - \nabla\mathcal{L}(\theta', \lambda')\|^2 \\
&= 2\|\nabla_\theta\mathcal{L}(\theta, \lambda) - \nabla_\theta\mathcal{L}(\theta, \lambda')\|^2 + 2\|\nabla_\lambda\mathcal{L}(\theta, \lambda) - \nabla_\lambda\mathcal{L}(\theta, \lambda')\|^2 \\
&\quad + 2\|\nabla_\theta\mathcal{L}(\theta, \lambda') - \nabla_\theta\mathcal{L}(\theta', \lambda')\|^2 + 2\|\nabla_\lambda\mathcal{L}(\theta, \lambda') - \nabla_\lambda\mathcal{L}(\theta', \lambda')\|^2.
\end{aligned}
$$

Bounding each term we have,

$$\|\nabla_\theta \mathcal{L}(\theta, \lambda) - \nabla_\theta \mathcal{L}(\theta, \lambda')\| \leq \left\| \sum_{j=1}^{J} (\lambda_j - \lambda'_j) \sum_{I \in \mathcal{J}_j} \sum_{k}^{K} \alpha_{j,I,k} \nabla_\theta P_k(D_I; \theta) \right\|_2$$
$$\leq c_1 \tau G_h \|\lambda - \lambda'\|_1$$
$$\leq c_1 \tau G_h \sqrt{J} \|\lambda - \lambda'\|_2.$$

$$\|\nabla_\lambda \mathcal{L}(\theta, \lambda) - \nabla_\lambda \mathcal{L}(\theta, \lambda')\| = 0.$$

$$\|\nabla_\theta \mathcal{L}(\theta, \lambda') - \nabla_\theta \mathcal{L}(\theta', \lambda')\| \leq \|\nabla_\theta \ell(\theta) - \nabla_\theta \ell(\theta')\| + \left\| \sum_{j=1}^{J} \lambda'_j \sum_{I \in \mathcal{J}_j} \sum_{k}^{K} \alpha_{j,I,k} (\nabla_\theta P_k(D_I; \theta) - \nabla_\theta P_k(D_I; \theta')) \right\|_2$$
$$\overset{(i)}{\leq} \beta_\ell \|\theta - \theta'\| + \|\Lambda\|_1 c_1 (G_h \tau + \tau^2 \beta_h) \|\theta - \theta'\|$$
$$\leq (\beta_\ell + c_1 \|\Lambda\|_1 \tau (G_h + \tau \beta_h)) \|\theta - \theta'\|$$

$$\|\nabla_\lambda \mathcal{L}(\theta, \lambda') - \nabla_\lambda \mathcal{L}(\theta', \lambda')\| \leq \left\| \sum_{j=1}^{J} \lambda'_j \sum_{I \in \mathcal{J}_j} \sum_{k=1}^{K} \alpha_{j,q,k} (P_k(h(\theta; x) - P_k(h(\theta'; x)))) \right\|$$
$$\overset{(ii)}{\leq} c_1 \tau \|\Lambda\|_1 G_h \|\theta - \theta'\|$$

Above, in $(i)$ we have used the fact that $P_k$ is the composition of a $G_h$-Lipschitz and $\beta_h$-smooth function with a $\tau$-Lipschitz and $\tau$-smooth function, resulting in a $(G_h \tau + \tau^2 \beta)$-smooth function. Similarly, in $(ii)$, we have used the fact that $P_k$ is the composition of two Lipschitz functions, resulting in another Lipschitz function.

Ultimately we obtain that $\|\nabla \mathcal{L}(\theta, \lambda) - \nabla \mathcal{L}(\theta', \lambda')\|^2$ is bounded by,

$$\max \left\{ (c_1 \tau G_h \sqrt{J})^2, [\beta_\ell + \|\Lambda\|_1 c_1 (G_h \tau + \tau^2 \beta_h)]^2 + [c_1 \tau \|\Lambda\|_1 G_h]^2 \right\} (\|\theta - \theta'\|^2 + \|\lambda - \lambda\|^2).$$

This implies that $f$ is $\beta$-smooth with $\beta = \beta_\ell + 2 c_1 \tau \cdot \max \left\{ G_h \sqrt{J}, \|\Lambda\|_1 (2 G_h + \tau \beta_h) \right\}$. $\qquad \square$

## E ADDITIONAL EXPERIMENTAL DETAILS

### E.1 DATASET AND PRE-PROCESSING DETAILS

We evaluate RaCO-DP on tabular fairness and privacy benchmark datasets from Lowy et al. (2023), namely, Adult (Becker & Kohavi, 1996), German Credit Card (Hofmann, 1994), and Parkinsons (Little, 2007). The classification task for Credit Card and Parkinsons is "whether the user will default payment the next month", and "whether the total UPDRS score of the patient is greater than the median or not," respectively. For Adult, the task is "whether the individual will make more than \$50K." In all tasks, the sensitive attribute is gender.

To evaluate on more diverse subgroups, we also evaluate RaCO-DP on folkstables which is the 2018 yearly American Community Survey. We use the python package 'folktables' (Ding et al., 2021) to download and process the data for the Alabama ("AL") state in the US. We choose a "5"-year horizon and choose survey option to be 'person.' We adopt the experimental setup of Lowy et al. (2023) (including classification task, pre-processing, etc.) and report the baselines results directly from the official repository (Gupta, 2023).

We also present results on the Heart Disease Health Indicators dataset Alex Teboul (21 risk factors).

To prove the scalability of RaCO-DP on neural network, we train a ResNet16 He et al. (2016) on the CelebA dataset. CelebA (CelebFaces Attributes Dataset) is a large-scale face dataset containing over

| Dataset | SGD | DP-SGD | RaCO-DP | DP-FERMI |
|---------|-----|--------|---------|----------|
| Adult | $0.018 \pm 0.001$ ms | $0.037 \pm 0.001$ ms | $0.064 \pm 0.010$ ms | $85 \pm 10$ ms |
| CreditCard | $0.020 \pm 0.005$ ms | $0.035 \pm 0.004$ ms | $0.055 \pm 0.003$ ms | $88 \pm 14$ ms |

Table 2: **Computational overhead comparison in terms of wall-time clock.**

200,000 celebrity images, each annotated with 40 attribute labels (e.g. smiling, eyeglasses, or hair color) and five landmark locations, widely used for training and testing in computer vision tasks such as face recognition. For the fairness constraints, CelebA is used with gender as the sensitive attribute.

### E.2 HYPERPARAMETER TUNING AND ACCOUNTING

Our hyperparameter selection process follows a two-phase approach. In the first phase, we run a hyperparameter search over predefined ranges: Gaussian noise variance $\sigma \in [3, 6]$, Laplace parameter $b \in [0.1, 0.5]$, learning rates $\eta_\theta, \eta_\lambda \in [10^{-4}, 0.1]$, mini-batch size $B \in [256, 1256]$, and softmax temperature $\tau \in [1, 10]$. We constrain the dual variables $\lambda$ to be non-negative by setting the projection set $\Lambda = (\mathbb{R}^+)^J$. For each configuration, we target a specific constraint value $\gamma$ and evaluate performance across five different seeds, selecting the hyperparameters that achieve the best validation accuracy while satisfying the constraint. In the second phase, we use the best hyperparameters identified through 200 optimization runs to train 20 new models. We report test accuracy and constraint satisfaction for each model based on the checkpoint that achieved the highest validation accuracy while satisfying the constraints on the train set.

Recent work by Lebeda et al. (2024) and Chua et al. (2024) shows that the common use of shuffled fixed-size mini-batches can violate guarantees from privacy accountants, which assume Poisson sampling. Therefore, we use Poisson sampling for the mini-batches.

### E.3 REGULARIZATION–PRIVACY–ACCURACY TRADE-OFFS

**Demographic Parity.** In Figure 4, we compare fairness–utility trade-offs of RaCO-DP against baseline methods on logistic regression models trained with demographic parity constraints across the `Adult`, `Credit-Card`, and `Parkinsons` datasets. We evaluate a range of privacy budgets $\varepsilon \in \{0.5, 1, 2, 9\}$. Consistent with the main results in Section 6, RaCO-DP significantly narrows the gap to the non-private baseline and Pareto-dominates existing private baselines.

**False Negative Rates.** Figure 5 illustrates the flexibility of our framework beyond fairness constraints by enforcing limits on false negative rate (FNR, i.e., $1 -$ recall). Controlling FNR is particularly important in medical settings. For example, on the `Heart` dataset, XGBoost achieves 90% accuracy but suffers from an FNR of 90%. Non-private SGDA reduces FNR to 58% with 87.5% accuracy, while DP-RaCO nearly matches this trade-off with 60% FNR at the same accuracy (Figure 5b). We also observe that once the constraint threshold is pushed beyond a certain point, RaCO-DP fails to further reduce FNR, an effect discussed in Section 6 (Limitations) and analyzed in Appendix E.6.

**Equalized Odds.** In Figure 6, we show results on the `Credit-Card` dataset for logistic regression trained under equalized odds constraints. The performance trends mirror those under demographic parity. Note that the DP-FERMI results are taken directly from Lowy et al. (2023) and not re-run; as reported, both mean and variance remain unchanged across privacy levels $\varepsilon \in \{0.5, 1, 3, 9\}$.

### E.4 HARD VS. SOFT CONSTRAINTS

Figure 8b compares soft and hard constraints for the dual update. Notably, the soft constraint (with tuned softmax temperature $\tau$) achieves similar performance compared to its hard constraint counterpart (solid dots) for most target values while maintaining similar levels of constraint satisfaction. This suggests that using soft constraints for the dual update does not significantly impact utility or constraint enforcement.

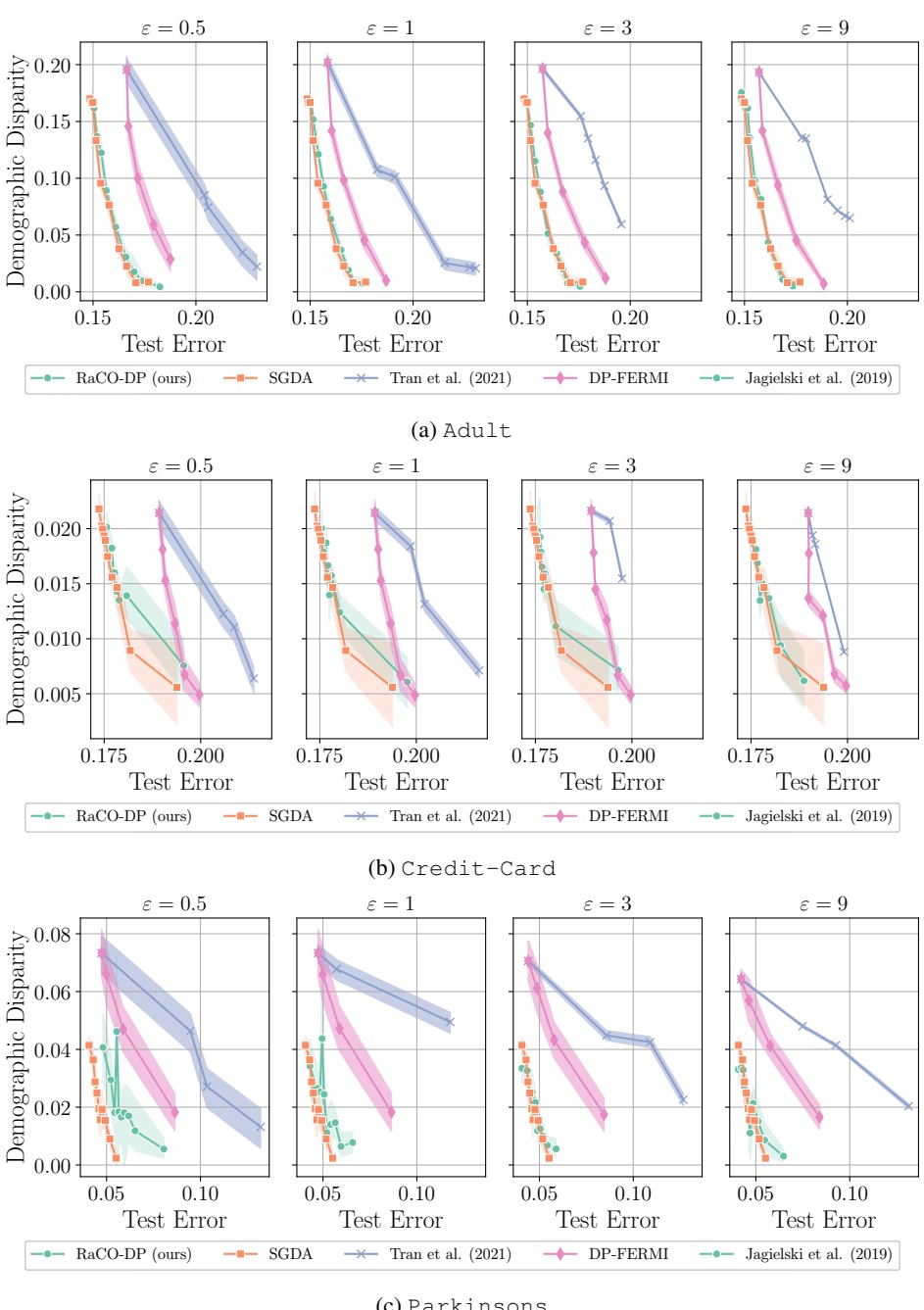

Figure 4: Fairness–utility trade-offs under **demographic parity constraints** for logistic regression on three benchmark datasets. RaCO-DP consistently reduces the gap to non-private performance and outperforms private baselines across privacy budgets $\varepsilon \in \{0.5, 1, 2, 9\}$.

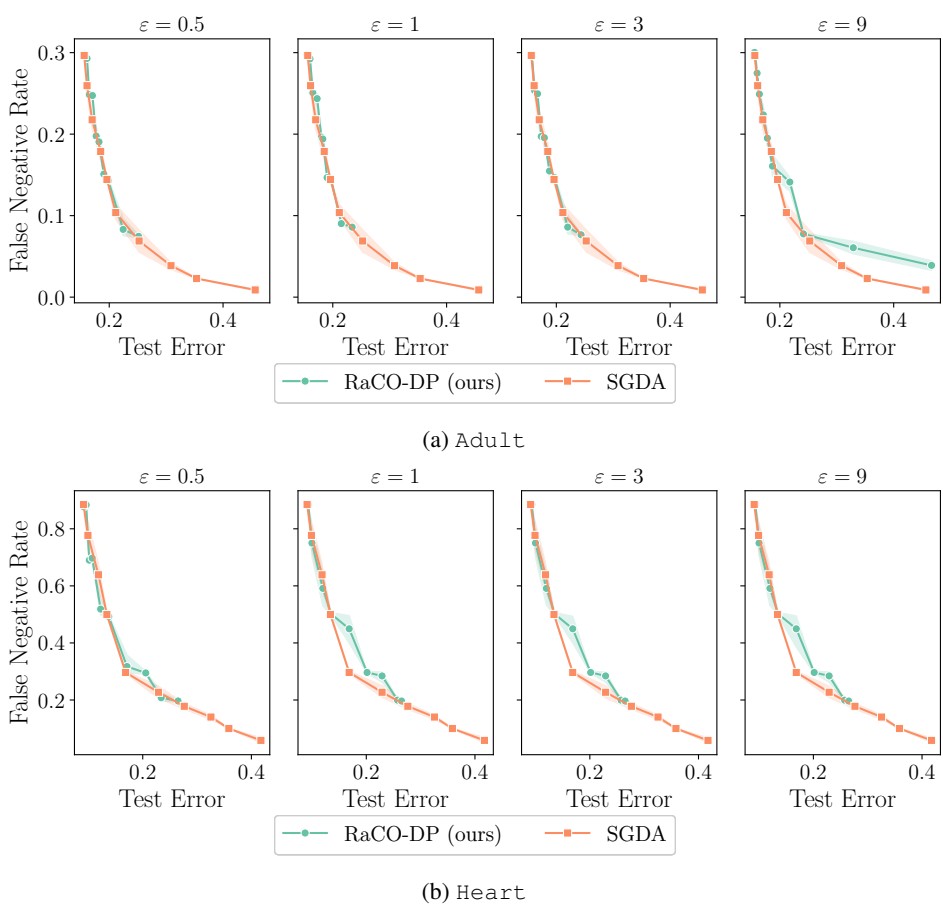

(a) `Adult`

(b) `Heart`

Figure 5: Performance–privacy trade-offs under a **false negative rate (FNR) constraint**. On the `Heart` dataset, RaCO-DP achieves accuracy–FNR trade-offs competitive with non-private baselines.

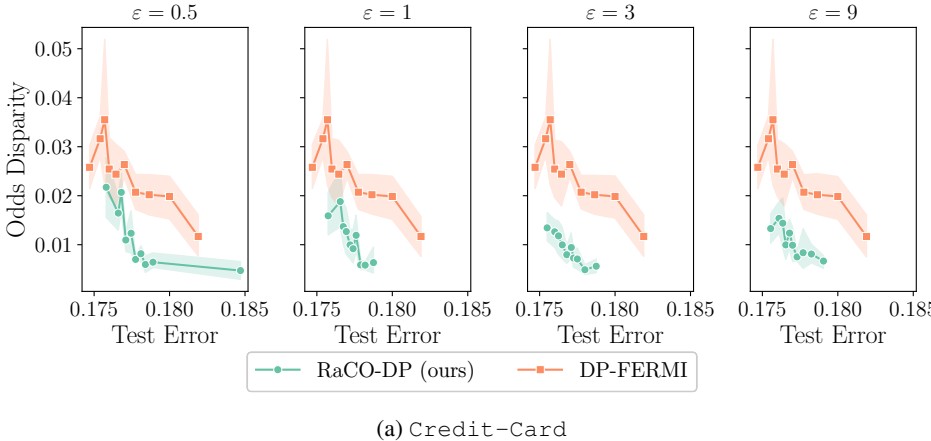

(a) `Credit-Card`

Figure 6: Fairness–utility trade-offs under **equalized odds constraints** for logistic regression on the `Credit-Card` dataset. Results for DP-FERMI are taken from Lowy et al. (2023).

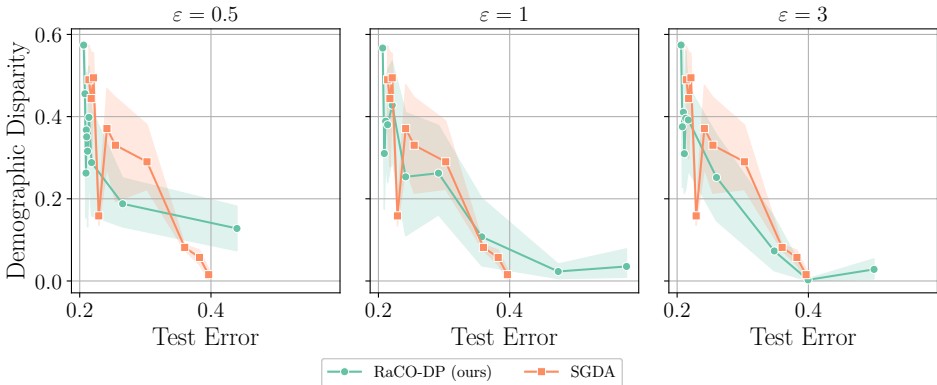

Figure 7: Fairness–utility trade-offs under **demographic parity** for logistic regression on the `ACSEmployment` dataset.

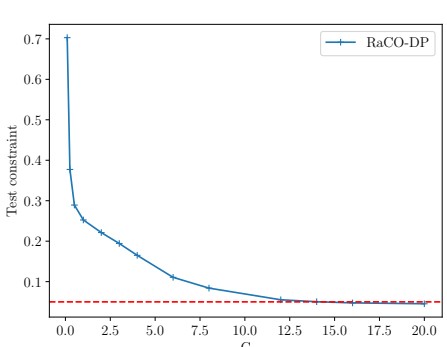

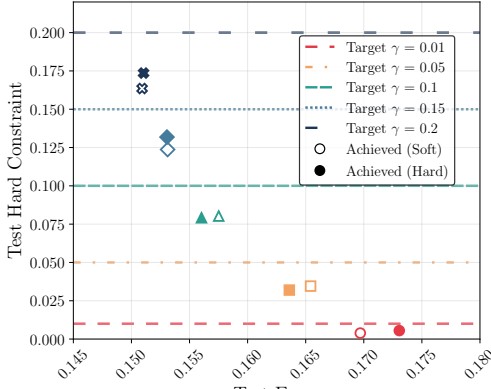

(a) **False Negative Rate-Constrained Classification on Adult.** The clipping norm $C$ plays a critical role in satisfying a pessimistic constraint ($\gamma = 0$), even without noise related to differential privacy ($\sigma = 0, b = \infty$).

(b) **Hard vs Soft Constraints on Adult.** Trade-off between test error and demographic parity. Dashed lines show target constraints, with soft (hollow circles) and hard (solid dots) implementations achieving similar performance.

Figure 8: **Constraint Analysis on Adult Dataset.** (a) Effect of clipping norm on false negative rate-constrained classification. (b) Comparison of hard vs. soft demographic parity constraints.

### E.5 COMPUTE PERFORMANCE

We provide a computational comparison between methods in Table 2, where we report the mean time of computing an SGD step, compared to a DP-SGD step and a RaCO-DP step on an eight-core CPU machine on Adult and Credit-Card on a batch size of 512. For reference, we also report the mean time of a DP-FERMI step using the publicly available implementation.

**RaCO-DP is 3 orders of magnitude faster to train than DP-FERMI on the same machine.** Our algorithm builds on DP-SGD with the only additional overhead being computing the dual updates, which scales linearly in the number of constraints.

We note that our method's extra cost over standard DP-SGD is computing the dual update, which, if implemented naively, scales linearly in the number of constraints $Q$, implying $Q$ extra backward passes to compute the gradients in the worst case. However, in practice, we can compute the gradient only for the active constraints ($\lambda_{(i)} > 0$), which can significantly reducing the computational costs.

### E.6 IMPACT OF THE CLIPPING NORM

Figure 8a shows how the clipping threshold $C$ affects RaCO-DP when we enforce a pessimistic constraint of FNR $< 0$ on the ADULT dataset, in a non-private setting ($\sigma = 0$, $b = \infty$). With a small clipping norm ($C \leq 2$) the empirical FNR violation is still above $0.6$, confirming that the bias introduced by clipping alone can drive the iterates far outside the feasible set. As the threshold increases, this bias shrinks rapidly; once $C \geq 12.5$, the FNR aligns with the target (red line), and the constraint is consistently satisfied.

These results demonstrate that an obstacle to satisfying the chosen constraints for our RaCO-DP is the bias from clipping and *not* the DP noise. Therefore, we stress that tuning $C$ is an important aspect when applying RaCO-DP in practice.

### E.7 DEEP LEARNING EXPERIMENT

We further evaluate RaCO-DP on deep neural networks by training a ResNet16 He et al. (2016) on CelebA under demographic parity constraints. Following common practice in private training Berrada et al. (2023), we replace batch normalization with group normalization (16 groups) to avoid reliance on batch statistics, which interact poorly with privacy constraints due to their reliance on batch-level statistics. Models are trained with a batch size of 256. For non-private baselines, we use a learning rate of $0.01$, while private models require a larger learning rate ($0.2$) to compensate for the effect of gradient clipping. For $\varepsilon = 1$ and $\varepsilon = 9$, we add Gaussian noise with $\sigma = 1.2$ and $\sigma = 0.6$, respectively, and set the Laplace noise scale $b = 0.5$. All results are averaged over five random seeds. Figure Figure 2a reports accuracy under varying disparity thresholds $\gamma \in \{1, 0.1, 0.075, 0.05\}$, highlighting that RaCO-DP maintains strong utility even under strict fairness and privacy requirements.

### E.8 EXPERIMENTS ON LOW EPSILON

In figure Figure 9 we employ RaCO-DP to train a logistic regression on Adult with demographic parity constraints at three restrictive privacy-budget settings: $\varepsilon \in \{0.01, 0.1, 1\}$. From a utility perspective, tightening the privacy budget from $\varepsilon = 1$ to $\varepsilon = 0.1$ results in only a modest decrease in accuracy of approximately three percentage points ($\approx 85\%$ to $\approx 82\%$). Even under a stringent budget of $\varepsilon = 0.01$, the most accurate model maintains an accuracy of about $80\%$. From a fairness perspective, smaller budgets lead to wider $95\%$ confidence intervals (CIs) for the disparity metric $\gamma$: the CI width increases from roughly $0.05$ at $\varepsilon = 0.1$ to about $0.10$ at $\varepsilon = 0.01$, with a mean $\gamma \approx 0.17$. The corresponding mean disparity gap between $\varepsilon = 0.01$ and $\varepsilon = 0.1$ can be as large as

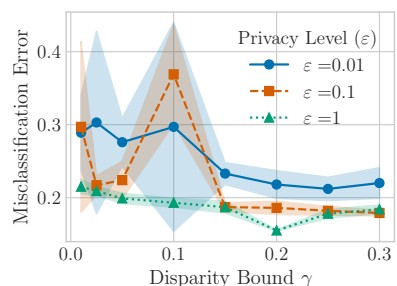

Figure 9: Variance in fairness–utility trade-offs at small privacy budgets.

$0.15$ ($15\%$), reflecting the greater noise introduced at lower $\varepsilon$. Overall, DP-RaCO exhibits graceful

degradation: while very small budgets ($\varepsilon = 0.01$) exacerbate the accuracy–fairness trade-off, a moderate budget ($\varepsilon = 0.1$) achieves reliable performance with a narrow CI ($\approx \pm 0.05$).

# F   BROADER IMPACT

An important takeaway from our work is that privacy and robustness criteria (such as the absence of performance disparities for underrepresented groups) are not inherently at odds with each other. This realization calls into question the practice of broadening the concept of privacy-utility trade-offs to include trade-offs with other robustness criteria. In high-risk decision-making systems that require both privacy and robustness, the responsibility for achieving such robustness falls on the beneficiaries of automated decision-making systems (governments and private institutions), as well as algorithm designers. These stakeholders must take care not to mistakenly attribute a lack of robustness to privacy mitigations, or a lack of privacy to robustness requirements.

Our work contributes to the existing literature in algorithmic fairness and privacy, and as such, adopts and further formalizes their computational interpretations of these human values. It is important to note that these interpretations, while useful in the contexts we have explored, are by no means collectively exhaustive. Specifically, the use of our algorithm does not ensure privacy in the broad sense, but rather in the limited sense of differential privacy, which protects the privacy of individuals whose data has been collected for training. Given the technical complexities of correctly implementing differential privacy, inappropriate tuning of model parameters or use outside its intended context can lead to a false sense of privacy—and, worse, may be exploited for privacy-washing by malicious actors in charge.

