# OpenReview forum: "Private Rate-Constrained Optimization with Applications to Fair Learning"
_ICLR.cc/2026/Conference — ICLR 2026 Poster_

### Official Review · Reviewer_sEj5 · 2025-10-27

**Soundness:** 3
**Presentation:** 3
**Contribution:** 3
**Rating:** 6
**Confidence:** 3

**Summary:**

This paper tackles the challenge of training machine learning models under rate-based constraints (e.g. group fairness metrics like demographic parity, equalized odds) while ensuring differential privacy (DP). The authors propose RaCO-DP, a differentially private stochastic gradient descent-ascent method for solving the Lagrangian of constrained optimization problems that involve group-level rate constraints. A key innovation is the use of generalized rate constraints, which unify various constraint types as computations on subgroup histograms. By privately estimating these subgroup statistics at each training step (with calibrated noise), the method enforces constraints with minimal extra privacy cost. The paper provides a rigorous theoretical analysis, proving that RaCO-DP converges to an approximate stationary point even for non-convex problems. Empirically, the approach is evaluated on multiple fairness-sensitive benchmarks (enforcing demographic parity, false negative rate limits, and equalized odds) across tabular datasets (Adult, Credit, Parkinsons, ACSEmployment) and a deep learning task (CelebA with ResNet16). Results show that RaCO-DP achieves state-of-the-art trade-offs between accuracy, fairness, and privacy – Pareto-dominating prior private fairness methods under the same privacy budgets. Notably, the method scales to scenarios with many subgroups (18 in one experiment) and maintains high utility even under stringent privacy (e.g. $\varepsilon=1$) on a deep neural network. In summary, this work introduces the first general DP framework for rate-constrained learning, demonstrating strong theoretical guarantees and improved privacy–utility–fairness performance in fair ML applications.

**Strengths:**

- Novelty and Generality: The paper bridges a significant gap by presenting the first general DP optimization framework for arbitrary rate constraints, beyond prior work that focused almost exclusively on fairness constraints. This general formulation means the method could apply to a wide range of constrained ML tasks (fairness, robustness, cost-sensitive learning) under DP, marking a clear advancement in differentially private optimization.

 - Algorithmic Innovation: The proposed RaCO-DP algorithm (DP-SGDA) cleverly overcomes the core challenge that rate-based constraints depend on global group statistics (violating per-sample DP assumptions). The solution – privately computing subgroup histograms each mini-batch – ensures constraints can be evaluated and optimized without leaking individual data, at a privacy cost equivalent to a histogram query per step. This design exploits the linear structure of the dual (Lagrange multiplier) updates to contain noise growth. It’s an elegant extension of DP-SGD to constrained problems, and the authors describe it clearly with a unified “generalized rate constraint” formulation and examples.

 - Theoretical Rigor: The paper provides a solid theoretical foundation. The authors prove convergence of RaCO-DP to an approximate stationary point even for non-convex objectives. The analysis is non-trivial – it accounts for the bias introduced by noisy gradient estimates and leverages the linear structure of the constraints to improve convergence speed. Such theoretical guarantees (with formal proofs in the appendix) lend credibility to the approach and are a notable strength, as prior DP-fairness works often lacked convergence analysis.

 - Empirical Performance: The method shows excellent empirical results on multiple benchmarks. Across standard tabular datasets, RaCO-DP consistently achieves higher accuracy for the same fairness level (or lower disparity for the same accuracy) compared to prior approaches like DP-FERMI. In other words, it Pareto-dominates the previous state-of-the-art in the accuracy–fairness–privacy trade-off. For example, on a deep learning task (CelebA, ResNet16), RaCO-DP reaches ~90% accuracy with only a 10% demographic parity gap under strong privacy ($\varepsilon=1$), which is very close to the non-private model’s 95% accuracy. These are impressive results, demonstrating that the privacy cost of fairness constraints can be kept low.

**Weaknesses:**

- Soft Constraint Enforcement: RaCO-DP uses a Lagrangian (dual ascent) approach, effectively treating constraints in a soft manner during training. While this is standard in constrained optimization, it means the model might violate constraints transiently or converge to a point that satisfies them only approximately (within the allowed slack $\gamma$). The authors themselves discuss that using “hard” constraint enforcement (always projecting onto the constraint set) would break their theoretical guarantees, and empirically gave only limited gains. This is a minor weakness in the sense that the final solutions do seem to meet the constraints to the desired degree, but strictly enforcing constraints at all times isn’t supported by the core algorithm. In scenarios where absolutely no constraint violation can be tolerated, additional measures might be needed (at the cost of privacy or convergence guarantees).

 - Hyperparameter Sensitivity (Clipping): The approach inherits the need for per-sample gradient clipping from DP-SGD, which can introduce bias. In constrained optimization, an improperly tuned clipping norm can even prevent satisfying the constraints. The authors note that clipping can push the solution outside the feasible set, making the choice of clip norm “a critical hyperparameter”. In practice, this likely requires careful tuning or expert knowledge to get right – a potential drawback for ease of use. The paper does illustrate this issue (showing that overly small clip norms fail a strict FNR=0 constraint) and acknowledges it as a general DP limitation, but it remains a practical challenge.

 - Scope of Experiments: While the method is formulated generally, the empirical evaluation is focused primarily on group fairness scenarios. All reported experiments enforce fairness-related constraints; other applications (like robust optimization or cost-sensitive learning mentioned in the motivation) are not demonstrated. This leaves a gap in showing the framework’s generality in practice. For example, it would strengthen the paper to see at least one non-fairness use-case or a discussion of how the method would handle a different type of rate constraint. As is, the results convincingly cover fairness, but the general claim is not fully backed by experiments.

**Questions:**

1. RaCO-DP enforces constraints up to a slack $\gamma$. In practice, if a user needs to guarantee a strict constraint ($\gamma$ very close to 0), what is the advised approach? The discussion in the paper suggests using hard constraints in the dual update breaks theory and gives limited benefit. Could the authors clarify the observed impact of using hard constraints: did it ever significantly improve fairness or was it truly negligible? Also, in scenarios like requiring zero disparity or error ($\gamma=0$), does RaCO-DP struggle due to noise? It would help to know if there’s an inherent limit to how tight a constraint can be made under DP noise before performance collapses (as hinted by the FNR=0 case).

2. The paper notes that the gradient clipping norm $C$ is critical and can affect the feasibility of constraints. How did you select the clipping norm in experiments, and how sensitive are the results to this choice? Is there a heuristic or adaptive strategy you can recommend for choosing $C$ to balance privacy (noise) and constraint satisfaction? Understanding this would help practitioners avoid the issue of failing to meet a constraint due to overly aggressive clipping.

3. The method privately computes a histogram of subgroup counts per batch. How does the privacy budget consumption scale with the number of groups or constraints? For instance, in the ACSEmployment case with 18 subgroups, was there a noticeable increase in noise or decrease in accuracy relative to cases with fewer groups? Some discussion on scaling to a larger number of constraints (or larger partitions $Q$) under DP would be insightful. For example, does each additional subgroup simply incur a small constant cost, or do variance and utility trade-offs worsen significantly?

---

> ### Author Response · Authors · 2025-11-20
>
> We thank the reviewer for their positive review, and we address the specific questions and areas of confusion below.
>
> ### **Hard vs. Soft Constraints**
> Our empirical results suggest that enforcing "hard" constraints offers negligible benefit over the soft Lagrangian approach. We also provide experiments in which we tune the softmax temperature $\tau$ and compare it with the hard rate constraints for the dual update in Figure 8b and Lines 1508-1511 of the Appendix. The soft constraint version (hollow circles) achieves nearly identical trade-offs to the hard constraint implementation (solid dots). The theoretical guarantees for the soft approach are stronger, making it the preferred choice in our experimental setup.
>
> ### **Strict Constraints and Noise**
> We found that the primary bottleneck in satisfying a “strict” constraint close to 0 is often gradient clipping bias rather than DP noise itself. In Figure 8a (Appendix E.6), we demonstrate that even with no DP noise, the algorithm fails to satisfy a strict False Negative Rate (FNR) constraint if the clipping norm $C$ is too small $(C < 2)$. The bias from clipping pushes the updates away from the feasible region. Once the clipping norm is sufficiently large $(C > 12.5)$, RaCO-DP satisfies the strict target. Indeed, selecting larger clipping norms would induce large noise, as we scale the variance of the Gaussian noise in the primal update with $C$.
>
> However, in practice, these “strict” constraints are rare, if not impossible. In terms of theoretical results, Cummings et al. 2019 [1] provide an impossibility result for DP learning and exact fairness; while also demonstrating a positive result for the case of approximate fairness (our settings). More recently, Mangold et al. 2023 [2] have shown that:
> > "the fairness (and accuracy) costs induced by privacy in differentially private classification vanishes at a $O(\sqrt{p}/n)$ rate, where $n$ is the number of training records, and $p$ the number of parameters."
>
> This is achieved through a proof of pointwise Lipschitz smoothness of group fairness metrics with respect to the model where the pointwise Lipschitz constant explicitly depends on the confidence margin of the model, and may be different for each sensitive group. Therefore, in the presence of minority groups in the data, we can expect a non-zero gap between private and non-private models.
>
> *[1] Cummings et al. "On the Compatibility of Privacy and Fairness." UMAP Adjunct, 2019.*
> *[2] Mangold et al. "Differential Privacy Has Bounded Impact on Fairness in Classification." ICML, 2023.*
>
> ### **Selecting the Clipping Norm**
> As detailed in Appendix E.2, we treated the clipping norm (similarly, learning rate and noise scaling) as a hyperparameter tuned via grid search on a validation set.
>
> ### **Scope of Experiments**
> We do present experiments with False Negative Rate (FNR) constraints (Figure 5, Appendix E.3) on both the adult and heart datasets, which represent performance constraints rather than a fairness one.
>
> ### **Scaling with Subgroups**
> The privacy guarantee (see Theorem 4.1) does not scale with the number of groups $Q$. As described in Section 4.1, we compute a histogram in which each sample contributes to exactly one partition $D_q$. Consequently, the L1 sensitivity of the histogram is always 1, regardless of $Q$.
>
> While the privacy cost is constant, the statistical error can increase if $Q$ is very large because the number of samples per subgroup could decrease (assuming the dataset size stays constant). In Theorem 5.2, we define $n = \min_{q \in [Q]}\{|D_q|\}$ as the size of the smallest subgroup and our convergence rate depends on the gradient error estimation error, which scales with $\tilde{O}(1/n)$ for the primal and dual updates, see lines 374-377 of Section 5 or Appendix D.5 for more details. Also, the convergence rate $\alpha$ depends on the rate constraint via the number of constraints $J$, the minimum partition size $n$ and the number of classes $K$, see for example the $\log(JKn/\rho)$ terms in Theorem D.1.
>
> In the ACSEmployment experiment (18 groups), we did not observe a collapse in performance. Figure 2c shows that RaCO-DP maintains a competitive utility-fairness trade-off even with this larger number of partitions, confirming that the method scales well as long as the smallest subgroup contains enough samples.

---

### Official Review · Reviewer_vscK · 2025-10-27

**Soundness:** 3
**Presentation:** 2
**Contribution:** 4
**Rating:** 6
**Confidence:** 2

**Summary:**

RaCO-DP is a novel framework for optimizing machine learning models under rate constraints with differential privacy, DP variant of SGDA.

**Strengths:**

- Formal convergence analysis of RaCO-DP,
- New SOTA claimed to be achieved on standard benchmarks for tabular datasets: CelebA, Parkinsons, ACSEmployment,
- Scales to multiple sensitive groups (tested up to 18),
- Scales beyond convex models to deep learning model (ResNet16 on CelebA),
- Stronger privacy guarantees than former approaches,
- Allows to specify directly maximum disparity.

**Weaknesses:**

- Results section is not clear, which exactly datasets displayed results achieving SOTA?
- Authors didn't run the experiments for SOTA but compared with published data from Lowy et al.

**Questions:**

Improve presentation of results adding details regarding which datasets RaCO-DP exactly achieved a new SOTA on.

---

> ### Author Response · Authors · 2025-11-20
>
> We thank the reviewer for their positive review, and we address the specific questions and areas of confusion below.
>
> ### **SOTA Datasets**
> We show improved fairness-accuracy trade-offs on every tabular dataset we tested:
> * On Parkinson's dataset (Figure 2.B) RaCo-DP beats the other privatized baselines DP-FERMI and Tran et al. (2021), producing about 3% lower error than the previous SOTA (DP-FERMI); and matching the non-private baseline (SGDA) for the first time for example for a tolerable fairness violation of $\gamma = 0.05$.
> * Figure 4 (in the Appendix E, Section E.4) shows similar results for Adult, Credit-Card.
>
> We should stress that achieving results close to the non-private baseline is the strongest possible result for any private algorithm.
>
> ### **Running Lowy et al.**
> For the results shared in the paper (Figure 2-C, Figure 4), we have indeed run and validated Lowy et al. using their dataloaders and seeds (using the publicly available repository [1]) and achieved the same results. The same dataloaders are used in the training of our models. Therefore, no data preprocessing step or batching disparity exists between the two methods. Matching the experimental setup (down to the dataloaders) allowed us to make comparisons to other non-SOTA results (such as Tran 2021, etc.), which were already beaten by DP-FERMI.
>
> *[1] https://github.com/devanshgupta160/Stochastic-Differentially-Private-and-Fair-Learning/*

---

### Official Review · Reviewer_AG8J · 2025-11-01

**Soundness:** 3
**Presentation:** 3
**Contribution:** 3
**Rating:** 6
**Confidence:** 3

**Summary:**

This paper proposes RaCO-DP, a DP variant of the stochastic gradient descent–ascent algorithm for solving the Lagrangian formulation of rate-constrained problems. This paper proposes an algorithm that enforces fairness and differential privacy simultaneously. They formulate “fairness” as a rate-constrained problem in empirical risk minimization. However, previous formulations apply only to binary classification tasks.

This paper proposes generalized rate constraints that can handle multi-class classification. For privately solving the ERM with generalized rate constraints, they propose RaCO-DP, a DP variant of the stochastic gradient descent–ascent algorithm. Specifically, they leverage the fact that the rate-constraint objective can be expressed as a function of a histogram of model predictions, and that this histogram is easy to privatize (its sensitivity is 1) with the Laplace mechanism. RaCO-DP first computes a private histogram of the predictions, and then combines it with DP-SGD–style updates in the descent and ascent steps.

Under standard assumptions in non-convex optimization, they prove that their algorithm achieves convergence to an approximate stationary point with a rate comparable to those in related minimization settings. They also provide experimental results showing that their algorithm achieves better fairness–accuracy trade-offs than previous algorithms across different privacy levels.

**Strengths:**

* The paper is clearly written and easy to read.
* It has significance as it generalizes prior fairness constraints that previously limited to the binary classification setting to the multiclass setting. It also proposes a novel DP algorithm that leverages a private histogram; empirically, the method achieves a better privacy–fairness trade-off, and the paper also provides a theoretical convergence analysis for the algorithm.
* It provides experimental investigations and ablations over different hyperparameter choices and privacy budgets. It also shows that it is computationally more efficient than prior work, with only ~2× per-step overhead compared with DP-SGD (and large speedups over DP-FERMI).

**Weaknesses:**

The method’s performance can degrade substantially when the smallest subgroup is tiny, because DP noise makes its histogram estimates inaccurate. The paper offers only limited discussion of this case.

**Questions:**

The convergence guarantee Theorem 5.2 does not depend on the rate constraint parameters. What are the dependencies?

How valid is the bounded dual space assumption (section 5)? Since a compact $\Lambda$ caps penalties, it can induce violations to the rate constraints.

---

> ### Author Response · Authors · 2025-11-20
>
> We thank the reviewer for their positive review, and we address the specific questions and areas of confusion below.
>
> ### **Performance for Small Subgroups**
> We agree that small subgroups present a standard challenge in differentially private learning. When a subgroup is small, the noise required to preserve privacy can overwhelm the signal from that group. This fundamental challenge has been well studied in the literature and is not unique to our method, see for example Xu et al. (2012) [1].
>
> While we do not develop on this when presenting our method in Section 4, our theoretical analysis in Section 5 explicitly quantifies this degradation. In Theorem 5.2, we define $n = \min_{q \in [Q]}\{|D_q|\}$ as the size of the smallest subgroup and our convergence rate depends on the gradient error estimation error, which scales with $\tilde{O}(1/n)$ for the primal and dual updates, see lines 374-377 of Section 5 or Appendix D.5 for more details. This confirms the reviewer's intuition: as $n \rightarrow 0$, the bound loosens, reflecting the difficulty of estimating histograms for tiny groups under DP.
>
> Finally, we would like to emphasize that the challenge of estimating rates over small subgroups is present even in the non-private case. For example, standard estimation lower bounds suggest we should not expect to estimate the demographic disparity rate for a subgroup to accuracy better than $1/\sqrt{n}$, where $n$ is the number of members in the subgroup.
>
> *[1] Xu et al. "Differentially Private Histogram Publication," ICDE, 2012.*
>
> ### **Dependencies in Theorem 5.2**
>
> For brevity, the informal statement in the main text suppresses constants, as stated in the theorem. The explicit dependencies are detailed in Theorem D.1 in Appendix D.2. The convergence rate $\alpha$ depends on the rate constraint via the number of constraints $J$, the minimum partition size $n$ and the number of classes $K$, see for example the $\log(JKn/\rho)$ terms in Theorem D.1. Other parameters, such as the $\alpha$ parameter, affect the convergence rate through their impact on the Lipschitz/Smoothness parameters of the loss. We refer to Lemma D.10 in Appendix D.7 for a more detailed analysis of how these quantities interact.
>
> ### **Bounded Dual Space**
>
> You are correct that bounding the dual space theoretically caps the maximum penalty applied to constraint violations. However, as noted in Section 5, finding an approximate stationary point for a non-convex min-max optimization problem is generally intractable without additional assumptions of this kind. For this reason, the assumption that the dual space is a compact convex set is standard in the analysis of non-convex-concave minimax optimization and is seen even in convex settings, see for example [1] and [2]. Experimentally, enforcing such a constraint is often not necessary to achieve good performance and constraint satisfaction. However, explaining this discrepancy is likely highly non-trivial, and far beyond the scope of this work.
>
> *[1] Agarwal et al. "A Reductions Approach to Fair Classification," 2018.*
> *[2] Lin et al. "On Gradient Descent Ascent for Nonconvex-Concave Minimax Problems," 2020.*

---

> > ### Comment · Reviewer_AG8J · 2025-11-27
> >
> > Thank you for your response. My concerns have been addressed

---

### Official Review · Reviewer_MESD · 2025-11-03

**Soundness:** 3
**Presentation:** 2
**Contribution:** 3
**Rating:** 8
**Confidence:** 3

**Summary:**

This work gives an algorithm for differentially private optimization in the setting of machine learning where we have "rate constraints," i.e., constraints that limit how the model predicts across different parts of the dataset. The main application is to fairness. As I understand it, no prior work considers this exact problem even non-privately, but we can compare with prior DP algorithms for important special instances. (The rate constraints here are adapted to multiclass classification and, unlike Cotter et al. (2019), use a differentiable notion of rate.)

Algorithmically, we solve a Lagrangian formulation of this problem with a DP version of stochastic gradient descent-ascent: we privately update the primal parameters, then the dual parameters, then we project them back to the feasible space. Before this, for each step, we run a private histogram to estimate the current model's predictions on the current mini-batch.

We get a theoretical convergence analysis (which seems to have to deal with several nontrivial obstacles) and experiments, where the proposed approach appears to essentially dominate prior work.

**Strengths:**

This seems like a very solid contribution. There's been a lot of work on DP optimization in both minmax and fairness settings, and this seems like it contains technical ideas that will be useful elsewhere. The algorithm will surely be an experimental baseline for future work.

**Weaknesses:**

I feel the presentation of the paper could be improved. Here are some areas I had particular difficulty:
1. The rate constraints here adapt the Cotter et al. (2019b) definition by extending it to multiclass, but also working with soft decisions. The latter is a major point in their paper and may be an important distinction in the context of fairness [see 1].
1. We also consider constraints that may look at overlapping parts of the dataset. I am still somewhat confused about this: am I given the rate constraints and then have to come up with the dataset partition myself? Does the manner in which I do this depend on the data?
1. Section 3: the example partition is Hispanic, Black, or Caucasian. This is actively confusing, since often "Hispanic" is treated as an ethnicity distinct from a race, so for example on the US Census one can be "Hispanic Black" or "Non-Hispanic White."
1. Section 4: what's the obstacle in obtaining the per-sample decomposition? What non-trivial operation is happening here?

These were the main issues I ran into when reading the paper. They caused a fair bit of difficulty in forming my evaluation of it.


[1] Canetti, Ran, et al. "From soft classifiers to hard decisions: How fair can we be?." Proceedings of the conference on fairness, accountability, and transparency. 2019.

**Questions:**

How are the rate constraints provided to Algorithm 1? Does the global partition depend on the rate constraints, or is it also supplied to the algorithm?

---

> ### Author Response · Authors · 2025-11-20
>
> We thank the reviewer for their positive review, and we address the specific questions and areas of confusion below.
>
> ### **Soft vs. Hard Rates**
> We primarily use soft rates to facilitate gradient-based optimization for the primal update, as noted in Lines 143-146. We also provide experiments in which we tune the softmax temperature $\tau$ and compare it with the hard rate constraints for the dual update in Figure 8b and Lines 1508-1511 of the Appendix. We stress that Cotter et al. (2023) [1] also tackle the soft relaxations. Notably, in Section 4.2, they state:
> > For example, we might consider replacing the indicators defining a rate with sigmoids, and then optimizing the Lagrangian. This solves the differentiability problem, but introduces a new one: a (mixed) Nash equilibrium would correspond to a solution satisfying the sigmoid-relaxed constraints, instead of the actual constraints. Interestingly, it turns out that we can seek to satisfy the original unrelaxed constraints, even while using a surrogate.
>
> *[1] Cotter et al.  “Optimization with Non-Differentiable Constraints with Applications to Fairness, Recall, Churn, and Other Goals.”  JMLR, 2019.*
>
> ### **Dataset Partitions and Rate Constraints**
>
> The user is expected to have a formulation of their rate constraint in the form of Eq. (6). Note, however, that these formulations are natural and widely used in their application domain. Table 1 in Appendix 1 lists three of these. And Cotter 2019, Table 3 lists even more.
>
> Let us demonstrate this in a concrete example. Imagine you have constraints for the overlapping groups of different genders and races (or other intersectionalities), the global partitioning is formed from the smallest intersectional subsets. For example, if you have two constraints: i) a demographic parity constraint for women of color, and ii) an additional fairness clause for equality of odds for working age groups, then here is the natural breakdown of the constraints' overlaps that informs the histogram formation:
> For demographic parity constraint (i), we need to look at a histogram of $\hat{Y} \times Z$ where $\hat{Y}=\{Predicted Class\}$ and $Z = \{Race Category\} \times \{Gender Category\}$.
> For equality of odds, we need to look at a histogram of size $\hat{Y} \times Y \times \tilde{Z}$ where $Y=\{Ground-Truth Class\}$ and $\tilde{Z} = \{Age Group\}$.
> Therefore, considering all these intersectionalities, the final histogram is going to be of size $Race \times Gender \times Age \times Predicted Class \times True Class$. This follows from the structure of the constraint—what fairness constraints are used–and also the data structure (schema)–in particular, how many categories exist per-feature.
>
> We emphasize that any algorithm attempting to solve such a constrained optimization problem has to account for these intersectionalities. What RaCO-DP does is to present a straightforward way to incorporate this natural structure (via the histogram formation) in the optimization procedure.
>
> ### **Per-Sample Decomposition**
> The difficulty arises because the Lagrangian term for the rate constraints is defined over the dataset/minibatch averages, not as a simple sum of independent individual losses as for standard losses used with DP-SGD. Standard DP-SGD requires clipping the gradient of the loss with respect to a single example $x_i$. However, our Lagrangian contains the term $R(\theta, \lambda; B)$ that depends on global information of the minibatch. We show in Lines 270-281 that this dependency is inevitable, but that we can lift it using the histogram $H$ in Lines 281-191. This decomposability issue has been observed in the literature before in other contexts, for example, contrastive losses for self-supervised learning (see, for example, [1]).
>
> *[1] Kong, et al. "Differentially Private Optimization for Non-Decomposable Objective Functions." arXiv:2310.03104, 2025.*
>
> ### **Terminology (Hispanic/Black/Caucasian)**
> We thank the reviewer for their attention to this point. Indeed, “Hispanic” is not a racial category in Census data. Figure 1 is an illustrative example and thus not based on real Census data. We have replaced “Hispanic” by “Asian” (which, as illustrated in the datasheet [1] and [2, Appendix B], is a census racial category) to avoid any confusion.
>
> *[1] Becker & Kohavi. "Adult." UCI Machine Learning Rep, 1996.*
>
> *[2] Ding et al. "Retiring Adult: New Datasets for Fair Machine Learning." arXiv:2108.04884, 2022.*

---

> > ### Comment · Reviewer_MESD · 2025-11-20
> >
> > Thank you for your response, it resolves all my questions. After reading it and the other reviews and responses, I retain my initial positive impression of the paper.

---

### Author Response · Authors · 2025-12-03
**Rebuttal Summary**

We thank all the reviewers for their constructive feedback and the area chair for their effort in conducting the discussion during this challenging period. To the best of our knowledge, our rebuttal and the revised paper have successfully addressed the concerns raised by the reviewers:

* **Reviewers MESD and AG8J** explicitly confirmed that their questions were resolved and their concerns addressed.
* We also provided detailed responses to **Reviewers vscK and sEj5**. No further concerns were raised by any reviewer following our revisions.

We highlight the main highlights of our work as recognized by the reviewers:

**Novelty & Generalization:**

1. **Reviewer sEj5** identified our work as the "first general DP framework for rate-constrained learning," bridging a significant gap in the literature.
2. **Reviewers MESD** and **AG8J** praised the method for generalizing prior binary-only constraints to the multiclass setting, with **MESD** noting that the paper contains "technical ideas that will be useful elsewhere" and the algorithm will serve as an "experimental baseline for future work."

**SOTA Empirical Performance:**

3. **Reviewer sEj5** noted that our method "Pareto-dominates" prior private fairness approaches.
4. **Reviewer vscK** highlighted that we achieved "New SOTA" on standard tabular benchmarks and successfully scaled to deep learning models (ResNet16 on CelebA).
5. **Reviewers vscK** and **sEj5** both commended the method's scalability, specifically its ability to handle a large number of sensitive groups without performance degradation.

**Theoretical Rigor:**

6. **Reviewers AG8J**, **vscK**, and **sEj5** all cited the formal convergence analysis as a key strength. **Reviewer sEj5** emphasized that providing such guarantees for non-convex objectives lends significant credibility, as this is often missing in prior DP-fairness work.

We hope these points demonstrate the significant value this paper offers to the community and justify a recommendation for acceptance.

Sincerely,
Authors

---

### Meta-Review · Area_Chair_BRTW · 2025-12-19

**Summary:**

This paper focuses on differentially private optimization in the presence of rate constraints (which allow to enforce group fairness). This generalizes prior fairness constraints limited to binary classification to the multiclass setting. The authors propose RaCO-DP, a differentially private version of stochastic gradient descent-ascent solving the Lagrangian formulation of the problem. The manuscript contains both a theoretical analysis of this algorithm (which overcomes several nontrivial technical obstacles) and convincing simulation experiments (achieving SOTA on tabular benchmarks and scaling, Pareto-dominating previous private fairness approaches, and scaling to deep learning models).

All reviewers agree this is a strong submission and there is clear consensus towards accepting the paper. I believe this will be a nice addition to ICLR's technical program.

**Reviewer Concerns:**

In short, all concerns of the reviewers appear to have been addressed in the thoughtful rebuttal provided by the authors. There is no significant concern that remains outstanding.

More precisely, Reviewer MESD commented on several passages which were unclear in the original manuscript: usage of soft vs. hard rates; dataset partition; Hispanic ethnicity; and the main obstacle in obtaining the per-sample decomposition. The reviewer later commented that all these issues were addressed in the rebuttal, and I concur.

Reviewer AG8J had concerns about the performance for small subgroups, the dependencies in Theorem 5.2 on rate constraint parameters and the validity of bounded dual space assumptions. The authors clarified these points, although some of the them remain (rather unavoidable) weaknesses.

Reviewer vscK requested clarifications on which datasets displayed results achieving SOTA and was concerned about the authors reporting results from Lowy et al., rather than running the experiments themselves. Both points were addressed in a way which I find satisfactory.

Finally, Reviewer sEj5 had questions about what happens when the slack $\gamma$ is small, about the selection of the clipping norm, and about the privacy budget consumption scaling with number of groups/constraints. Again, I find that these issues have been vastly resolved by the rebuttal.

**Reviewer Scores:**

There is already consensus towards acceptance and reviewers' concerns have been properly addressed.

---

### Decision · Program_Chairs · 2026-01-26

Accept (Poster)